EMBO
Molecular Medicine

# Rapamycin rescues mitochondrial myopathy via coordinated activation of autophagy and lysosomal biogenesis

Gabriele Civiletto[1], Sukru Anil Dogan[1], Raffaele Cerutti[1], Gigliola Fagiolari[2], Maurizio Moggio[2], Costanza Lamperti[3], Cristiane Benincá[1], Carlo Viscomi[1,*] & Massimo Zeviani[1,**]

## Abstract

The mTOR inhibitor rapamycin ameliorates the clinical and biochemical phenotype of mouse, worm, and cellular models of mitochondrial disease, via an unclear mechanism. Here, we show that prolonged rapamycin treatment improved motor endurance, corrected morphological abnormalities of muscle, and increased cytochrome c oxidase (COX) activity of a muscle-specific Cox15 knockout mouse ($Cox15^{sm/sm}$). Rapamycin treatment restored autophagic flux, which was impaired in naïve $Cox15^{sm/sm}$ muscle, and reduced the number of damaged mitochondria, which accumulated in untreated $Cox15^{sm/sm}$ mice. Conversely, rilmenidine, an mTORC1-independent autophagy inducer, was ineffective on the myopathic features of $Cox15^{sm/sm}$ animals. This stark difference supports the idea that inhibition of mTORC1 by rapamycin has a key role in the improvement of the mitochondrial function in $Cox15^{sm/sm}$ muscle. In contrast to rilmenidine, rapamycin treatment also activated lysosomal biogenesis in muscle. This effect was associated with increased nuclear localization of TFEB, a master regulator of lysosomal biogenesis, which is inhibited by mTORC1-dependent phosphorylation. We propose that the coordinated activation of autophagic flux and lysosomal biogenesis contribute to the effective clearance of dysfunctional mitochondria by rapamycin.

**Keywords** autophagy; lysosomal biogenesis; mitochondrial disease; mTORC1; rapamycin

**Subject Categories** Genetics, Gene Therapy & Genetic Disease; Musculoskeletal System

## Introduction

Mitochondrial diseases are genetically and clinically heterogeneous conditions characterized by defects of oxidative phosphorylation (OXPHOS). Mutations in either nuclear or mitochondrial DNA (mtDNA) genes can lead to mitochondrial dysfunction, which may result in a wide array of clinical manifestations, ranging from relatively benign, late-onset (encephalo)myopathies to extremely severe, early-onset encephalopathies, cardiomyopathies, or multisystem failure (Gorman et al, 2016). No cure or effective therapy is currently available for mitochondrial diseases, although recent studies showed potential benefit of several approaches, at least in preclinical in vivo models (Chinnery, 2015; Viscomi et al, 2015; Viscomi, 2016). Given the clinical, biochemical, and genetic heterogeneity of mitochondrial diseases, and the rarity of the single syndromes, therapeutic approaches with wide applicability would be preferable, but this is a difficult task to achieve (Viscomi et al, 2015).

Rapamycin is a widely used inhibitor of mechanistic target of rapamycin (mTOR), a cytosolic Ser/Thr kinase belonging to the phosphatidylinositol kinase-related family of protein kinases, with central roles in several cellular processes, including protein translation, immune response, nucleotide and lipid synthesis, glucose metabolism, autophagy, and lysosomal biogenesis (Saxton & Sabatini, 2017). In the cytosol, mTOR forms two complexes, mTORC1 and mTORC2. The main function of mTORC1 is the activation of anabolic processes, whereas mTORC2 has major roles in regulating the structure of cytoskeleton and cell proliferation. Rapamycin acts by forming a ternary complex with the peptidyl-prolyl-isomerase FK506-binding protein 12 (FKBP12), a component of mTORC1 (Saxton & Sabatini, 2017), and the FKBP12-rapamycin binding (FRB) domain of mTOR. This prevents substrate recruitment at the active site (Yang et al, 2013), leading to a block of mTORC1, and a switch of cell metabolism toward catabolic processes. By contrast, mTORC2 is insensitive to acute rapamycin exposure, although prolonged treatment does also abrogate mTORC2 signaling, as

1 MRC Mitochondrial Biology Unit, University of Cambridge, Cambridge, UK
2 Neuromuscular and Rare Diseases Unit, Fondazione IRCCS Ca' Granda Ospedale Maggiore Policlinico, Milan, Italy
3 IRCCS Foundation Neurological Institute "C. Besta", Milano, Italy
*Corresponding author. Tel: +44 1223 252804; Fax: +44 1223 252715; E-mail: cfv23@mrc-mbu.cam.ac.uk
**Corresponding author. Tel: +44 1223 252700; Fax: +44 1223 252715; E-mail: mdz21@mrc-mbu.cam.ac.uk

rapamycin-bound mTOR cannot be incorporated into new mTORC2 complexes (Saxton & Sabatini, 2017). One of the consequences of rapamycin inhibition of mTORC1 is the activation of (macro)autophagy. This is a highly conserved process conveying substrates, including protein aggregates, infective agents and organelles, to the lysosome for degradation. Targets for degradation are first enclosed into specialized, double-membrane structures called autophagosomes, whose formation (phagophore), elongation, and closure are controlled by autophagy-related (ATG) proteins (Bento *et al*, 2016; Menzies *et al*, 2017). Autophagosomes are eventually fused with lysosomes to generate autophagolysosomes that carry out the digestion of the substrates. mTOR-dependent and mTOR-independent pathways regulate autophagy. The former involves the activation of class I phosphoinositide 3-kinase (PI3K), protein kinase B (PKB), and mTORC1, which blocks autophagy by phosphorylating ULK1, a kinase driving autophagosome formation, thus preventing its activation by AMPK. The mTOR-independent pathway involves numerous targets and can include the activation of the cAMP-dependent exchange protein (EPAC), the small G protein RAP2B, and phospholipase C epsilon (PLCε). PLCε increases the production of inositol-1,4,5-triphosphate and, as a consequence, the release of calcium from the endoplasmic reticulum. This in turn leads to the activation of the calcium-dependent phosphatase calpain, which inhibits autophagy.

Rapamycin has recently been reported as beneficial in mouse and cellular models of mitochondrial disease (Johnson *et al*, 2013, 2015; Zheng *et al*, 2016; Felici *et al*, 2017; Khan *et al*, 2017; Siegmund *et al*, 2017). The mechanisms underlying these effects are poorly understood. Although activation of autophagy upon rapamycin treatment has been reported in some of these studies (Johnson *et al*, 2015; Khan *et al*, 2017), others failed to detect it (Felici *et al*, 2017; Siegmund *et al*, 2017), but all were based on the measurement of autophagy markers at the steady state, whereas autophagic flux was not investigated in detail.

Here, we used the $Cox15^{sm/sm}$ mouse (Viscomi *et al*, 2011; Civiletto *et al*, 2015), in order to investigate how mTORC1 signaling and autophagy flux are regulated in a skeletal muscle-restricted mitochondrial disease model, and whether and how rapamycin can modify these mechanisms and impact on the myopathic phenotype. Furthermore, we compared the effects of rapamycin, an mTORC1-dependent autophagy activator, with those of rilmenidine, a pro-autophagic drug acting through the mTORC1-independent pathway.

## Results

### Rapamycin improves motor endurance, muscle morphology, and mitochondrial structure of $Cox15^{sm/sm}$ mice

$Cox15^{sm/sm}$ mice lack the gene encoding Cox15 in skeletal muscle (Viscomi *et al*, 2011; Civiletto *et al*, 2015). Cox15 oxidizes the C8-methyl group of heme O during the biosynthesis of the cytochrome c oxidase (cIV, COX)-specific heme A. $Cox15^{sm/sm}$ mice are characterized by profound COX deficiency, leading to severe myopathy. We administered rapamycin intraperitoneally (i.p.) (8 mg/kg/day) to both $Cox15^{sm/sm}$ and wild-type (*WT*) weaning littermates (21 days after birth), for 4 weeks. Rapamycin induced

a mild, non-significant growth retardation in both treated *WT* and $Cox15^{sm/sm}$ animals (not shown), and a significant improvement of the motor performance, measured by standard treadmill test, from week 2 to the end of the treatment (Fig 1A). Histological and histochemical analysis of skeletal muscle revealed a consistent amelioration of pathology in rapamycin-treated versus vehicle-treated (naïve) $Cox15^{sm/sm}$ animals (Fig 1B). Hematoxylin and eosin (H&E) staining showed a significant increase in the cross-sectional area (CSA) of the muscle fibers (Fig 1C) and a drastic reduction in centralized nuclei (Fig 1D) in rapamycin-treated versus naïve $Cox15^{sm/sm}$ mice. PAS staining also revealed an accumulation of glycogen in skeletal muscle of naïve $Cox15^{sm/sm}$ mice, which was further and significantly increased in treated $Cox15^{sm/sm}$ animals (Fig 1E). COX histochemistry in skeletal muscle showed significantly increased COX reaction (Fig 1F), and a parallel reduction of SDH reaction, another index of mitochondrial proliferation, in treated versus naïve $Cox15^{sm/sm}$ animals (Fig 1G). Spectrophotometric measurements in muscle homogenates (Fig 2A) showed that citrate synthase (CS) activity, an index of mitochondrial mass, was significantly increased in naïve $Cox15^{sm/sm}$, and returned to normal values in muscle homogenates of rapamycin-treated $Cox15^{sm/sm}$ mice. The specific activity of COX (cIV) was significantly lower in $Cox15^{sm/sm}$ samples and remained unchanged under rapamycin treatment. As a consequence, cIV/CS activity increased in treated versus naïve $Cox15^{sm/sm}$ (Fig 2A). The protein levels of NDUFA9, a cI subunit; UQCRC2, a cIII subunit; and ATP5A, a cV subunit, were markedly increased in naïve $Cox15^{sm/sm}$ versus *WT* and were reduced to *WT* levels upon rapamycin treatment (Fig 2B), while cII subunit SDHA was unchanged in untreated $Cox15^{sm/sm}$ versus *WT* and was slightly reduced after rapamycin treatment. cIV subunit COX4 was profoundly reduced in both rapamycin-treated and untreated $Cox15^{sm/sm}$ mice, as expected given the role of COX15 in cIV biogenesis. No differences were detected between treated versus naïve *WT* animals. One-dimension blue native gel electrophoresis (1D-BNGE) in-gel activity and immunovisualization confirmed a slight increase in COX reaction and amount of fully assembled cIV in rapamycin-treated versus naïve $Cox15^{sm/sm}$ muscle samples (Fig 2C and D), while no differences were detected in complex I (not shown). Pyruvate/glutamate/malate (not shown)- and succinate-stimulated oxygen consumption (Fig 2E), which was significantly reduced in $Cox15^{sm/sm}$ versus *WT* mitochondria isolated from skeletal muscles, again significantly increased in treated $Cox15^{sm/sm}$ animals (Fig 2E).

Upon rapamycin treatment, mtDNA copy number in skeletal muscle, which was increased in naïve $Cox15^{sm/sm}$ mice, returned to levels comparable to *WT* (Fig 2F).

Modified Gomori trichrome staining showed a remarkable decrease in the reddish hue of the muscle fibers, indicating reduced mitochondrial proliferation in treated versus naïve $Cox15^{sm/sm}$ mice. A decrease in "blue fibers", i.e., fibers completely devoid of COX reaction, was also detected in treated versus naïve $Cox15^{sm/sm}$ (Appendix Fig S1).

Ultrastructurally, while a huge number of giant mitochondria with big vacuoles and disrupted cristae were detected in naïve $Cox15^{sm/sm}$ muscle fibers, damaged mitochondria were significantly decreased in the rapamycin-treated $Cox15^{sm/sm}$ samples (Fig 3 and Appendix Fig S2). Quantitative EM analysis confirmed a significant

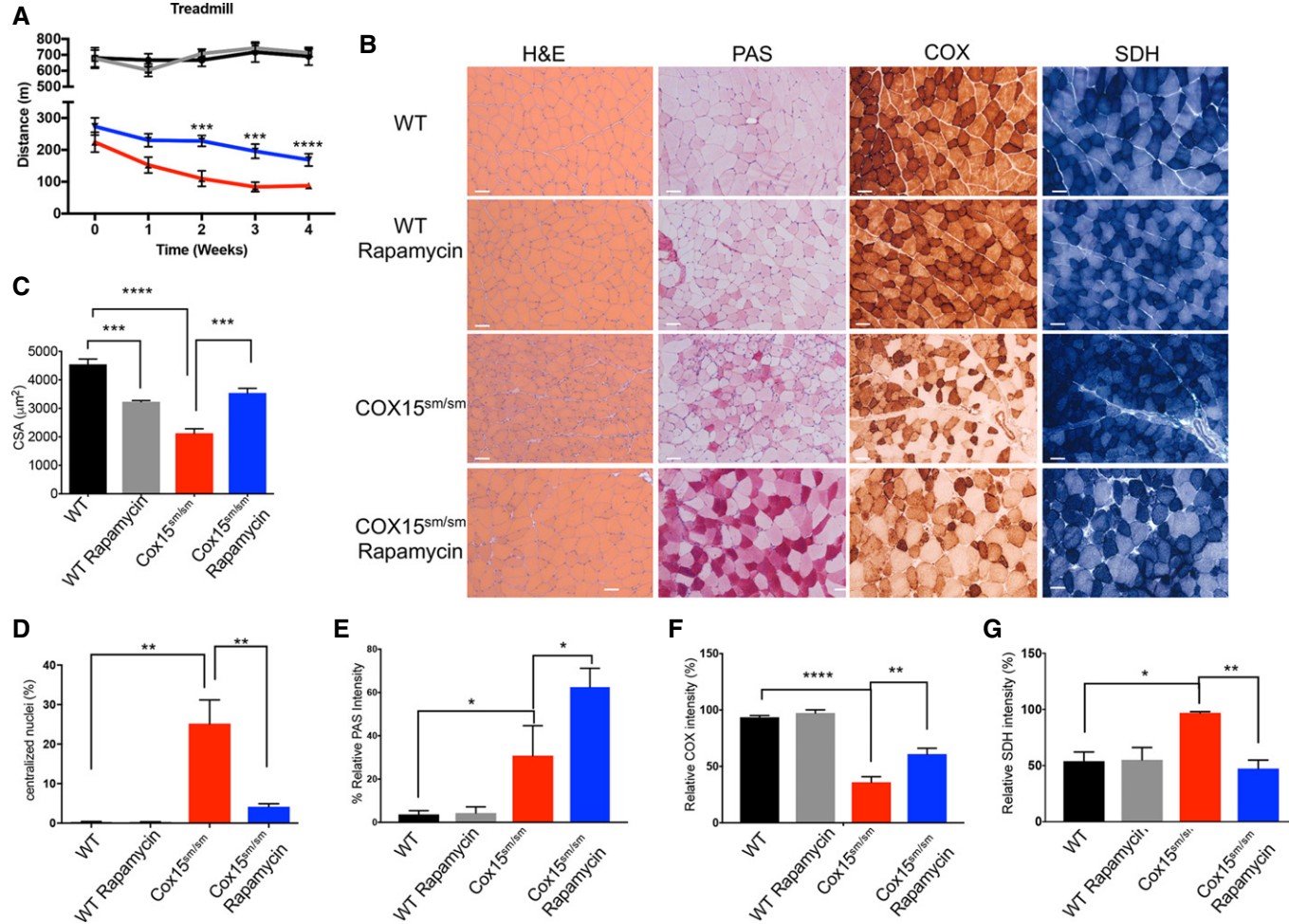

**Figure 1.  Rapamycin treatment improves the myopathic phenotype of Cox15$^{sm/sm}$ mice.**

A   Treadmill analysis (n = 8/group). Black: untreated WT; gray: rapamycin-treated WT; red: untreated Cox15$^{sm/sm}$; blue: rapamycin-treated Cox15$^{sm/sm}$. Error bars represent SEM. The asterisks represent the significance levels calculated by unpaired, two-tailed Student's t-test: ***P = 0.005 (week 2); ***P = 0.0007 (week 3); and ****P < 0.0001 (week 4). Only P-values for rapamycin-treated versus untreated Cox15$^{sm/sm}$ mice are shown.

B   Histological and histochemical characterization of skeletal muscle in rapamycin-treated and untreated Cox15$^{sm/sm}$ and WT mice. H&E: hematoxylin and eosin; PAS: periodic acid–Schiff reaction; COX: cytochrome c oxidase; SDH: succinate dehydrogenase. Scale bars correspond to 50 μm.

C   Analysis of the cross-sectional area of muscle fibers in the different genotypes (n = 3/group). Black: untreated WT; gray: rapamycin-treated WT; red: untreated Cox15$^{sm/sm}$; blue: rapamycin-treated Cox15$^{sm/sm}$. Error bars represent SEM. The asterisks represent the significance levels calculated by one-way ANOVA with Tukey's post hoc multiple comparison test: ***P = 0.001; ***P = 0.006; and ****P < 0.0001.

D   Analysis of the number of centralized nuclei in muscle fibers (n = 3/group). Black: untreated WT; gray: rapamycin-treated WT; red: untreated Cox15$^{sm/sm}$; blue: rapamycin-treated Cox15$^{sm/sm}$. Error bars represent SEM. The asterisks represent the significance levels calculated by one-way ANOVA with Tukey's post hoc multiple comparison test: **P = 0.0018 (Cox15$^{sm/sm}$ versus WT), **P = 0.0052 (Cox15$^{sm/sm}$-rapamycin versus Cox15$^{sm/sm}$).

E   Analysis of PAS-reaction intensity in muscle fibers (n = 4/group). Black: untreated WT; gray: rapamycin-treated WT; red: untreated Cox15$^{sm/sm}$; blue: rapamycin-treated Cox15$^{sm/sm}$. Error bars represent SEM. The asterisks represent the significance levels calculated by one-way ANOVA with Tukey's post hoc multiple comparison test: *P = 0.0293 (WT versus Cox15$^{sm/sm}$) and *P = 0.0150 (Cox15$^{sm/sm}$ versus Cox15$^{sm/sm}$+rapamycin).

F   Analysis of COX-reaction intensity in muscle fibers (n = 3). Black: untreated WT; gray: rapamycin-treated WT; red: untreated Cox15$^{sm/sm}$; blue: rapamycin-treated Cox15$^{sm/sm}$. Error bars represent SEM. The asterisks represent the significance levels calculated by one-way ANOVA with Tukey's post hoc multiple comparison test: ****P < 0.0001; **P = 0.0031.

G   Analysis of SDH-reaction intensity in muscle fibers (n = 3). Black: untreated WT; gray: rapamycin-treated WT; red: untreated Cox15$^{sm/sm}$; blue: rapamycin-treated Cox15$^{sm/sm}$. Error bars represent SEM. The asterisks represent the significance levels calculated by one-way ANOVA with Tukey's post hoc multiple comparison test: *P = 0.0199, **P = 0.0091.

decrease in damaged mitochondria upon rapamycin treatment, whereas the number of autophaged mitochondria, i.e., organelles engulfed in autophagic vacuoles, was comparable in rapamycin-treated versus naïve Cox15$^{sm/sm}$ (Appendix Fig S2). No differences were observed in the number of mitochondria with normal structure.

Altogether, these data indicate that rapamycin treatment remarkably ameliorates the myopathic phenotype, possibly through

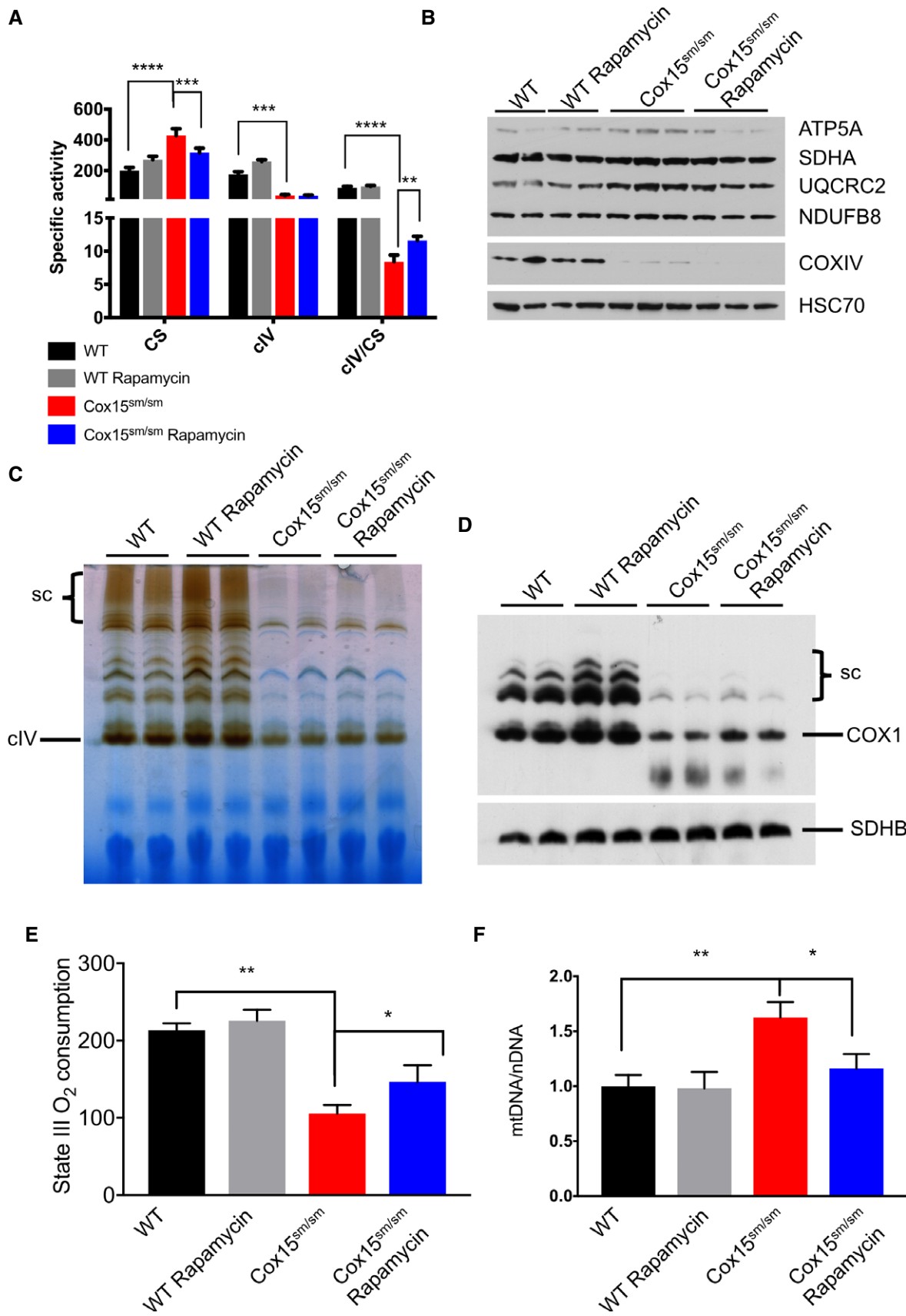

Figure 2.

**Figure 2.   Rapamycin improves the biochemical phenotype of *Cox15^sm/sm* skeletal muscles.**

A   Spectrophotometric activities of the respiratory chain (*n* = 4–5/group). CS: citrate synthase; cIV: complex IV. Black: untreated *WT*; gray: rapamycin-treated *WT*; red: untreated *Cox15^sm/sm*; blue: rapamycin-treated *Cox15^sm/sm*. Error bars represent SEM. The asterisks represent the significance levels calculated by two-way ANOVA with Tukey's correction: ****$P < 0.0001$ (CS: WT versus *Cox15^sm/sm*), ***$P = 0.0010$ (CS: *Cox15^sm/sm* versus *Cox15^sm/sm* rapamycin), ***$P = 0.005$ (cIV: *Cox15^sm/sm* versus WT), ****$P < 0.0001$ (cIV/CS: WT versus *Cox15^sm/sm*), **$P = 0.0060$ (cIV/CS: *Cox15^sm/sm* rapamycin versus *Cox15^sm/sm*).

B   Western blot immunovisualization of subunits of the respiratory complexes. Note the increased protein levels in *Cox15^sm/sm* versus *WT* samples, which are reduced to normal levels upon rapamycin treatment. Additional samples were run on a separate gel (not shown).

C   BNGE in-gel activity for cIV. Sc: supercomplexes. Note that the COX reaction is slightly increased in rapamycin-treated versus untreated *Cox15^sm/sm* samples.

D   Immunoblot of 1D-BNGE using an anti-cIV antibody (COX1). Note that COX amount is slightly increased in rapamycin-treated versus untreated *Cox15^sm/sm* muscles. SDHB was used as a loading control.

E   State III (succinate-driven) oxygen consumption (*n* = 3/group). Error bars represent SEM. The asterisks represent the significance levels calculated by unpaired, two-tailed Student's *t*-test: **$P = 0.0016$ (WT versus *Cox15^sm/sm*); *$P = 0.047$ (*Cox15^sm/sm* versus *Cox15^sm/sm*+rapamycin).

F   Analysis of mtDNA amount (*n* = 9/group). Error bars represent SEM. The asterisks represent the significance levels calculated by one-way ANOVA with Tukey's correction: **$P = 0.0020$; *$P = 0.0296$.

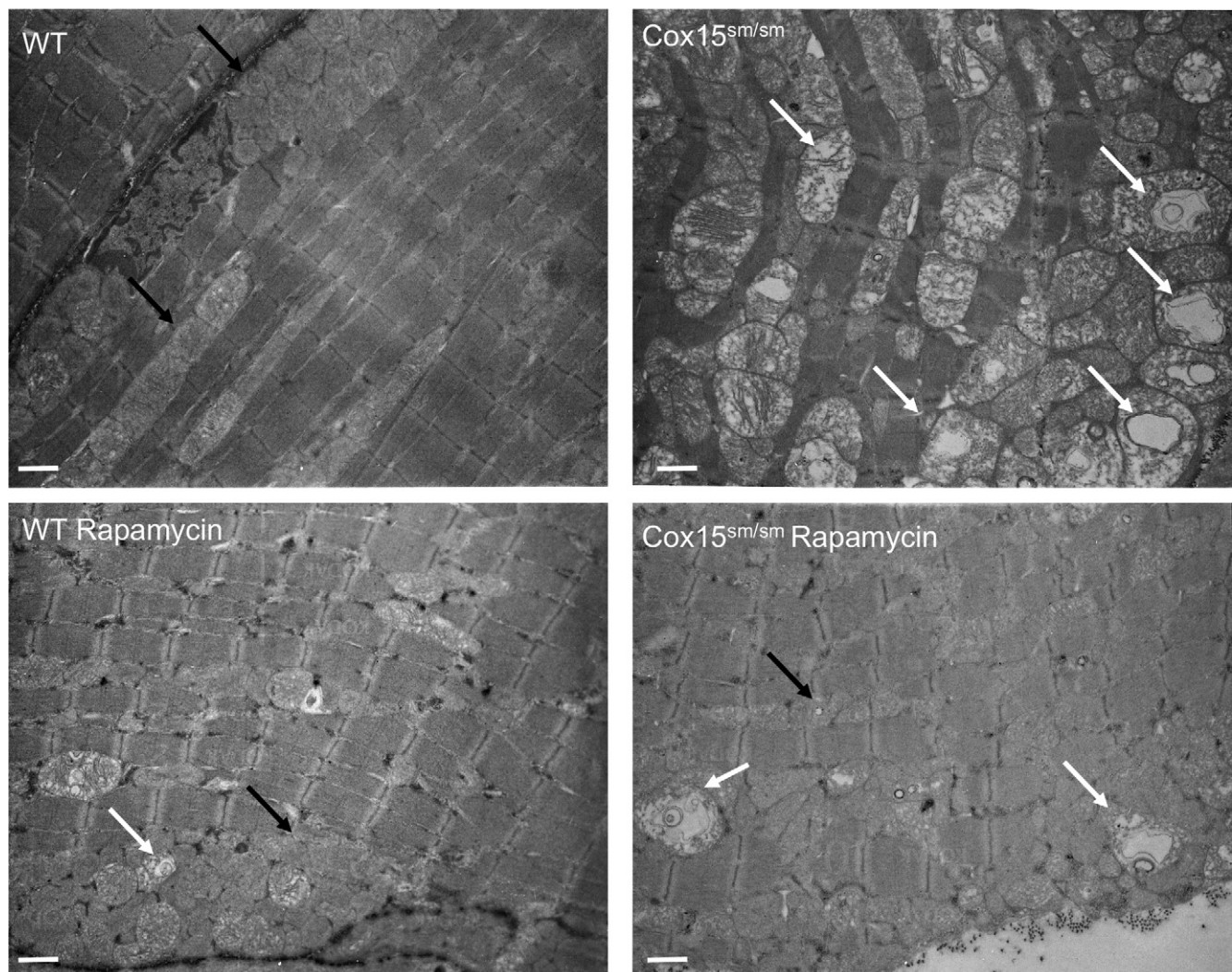

**Figure 3.   Rapamycin improves mitochondrial ultrastructure in *Cox15^sm/sm* mice.**

Note the accumulation of profoundly altered mitochondria in *Cox15^sm/sm* muscles (white arrows), possibly as a result of partially digested organelles within endolysosomes. Black arrows indicate examples of normal mitochondria. In rapamycin-treated *Cox15^sm/sm* animals, only a few of these structures are present, which can be detected also in rapamycin-treated *WT* samples. The scale bar corresponds to 1,136 nm (4,400×).

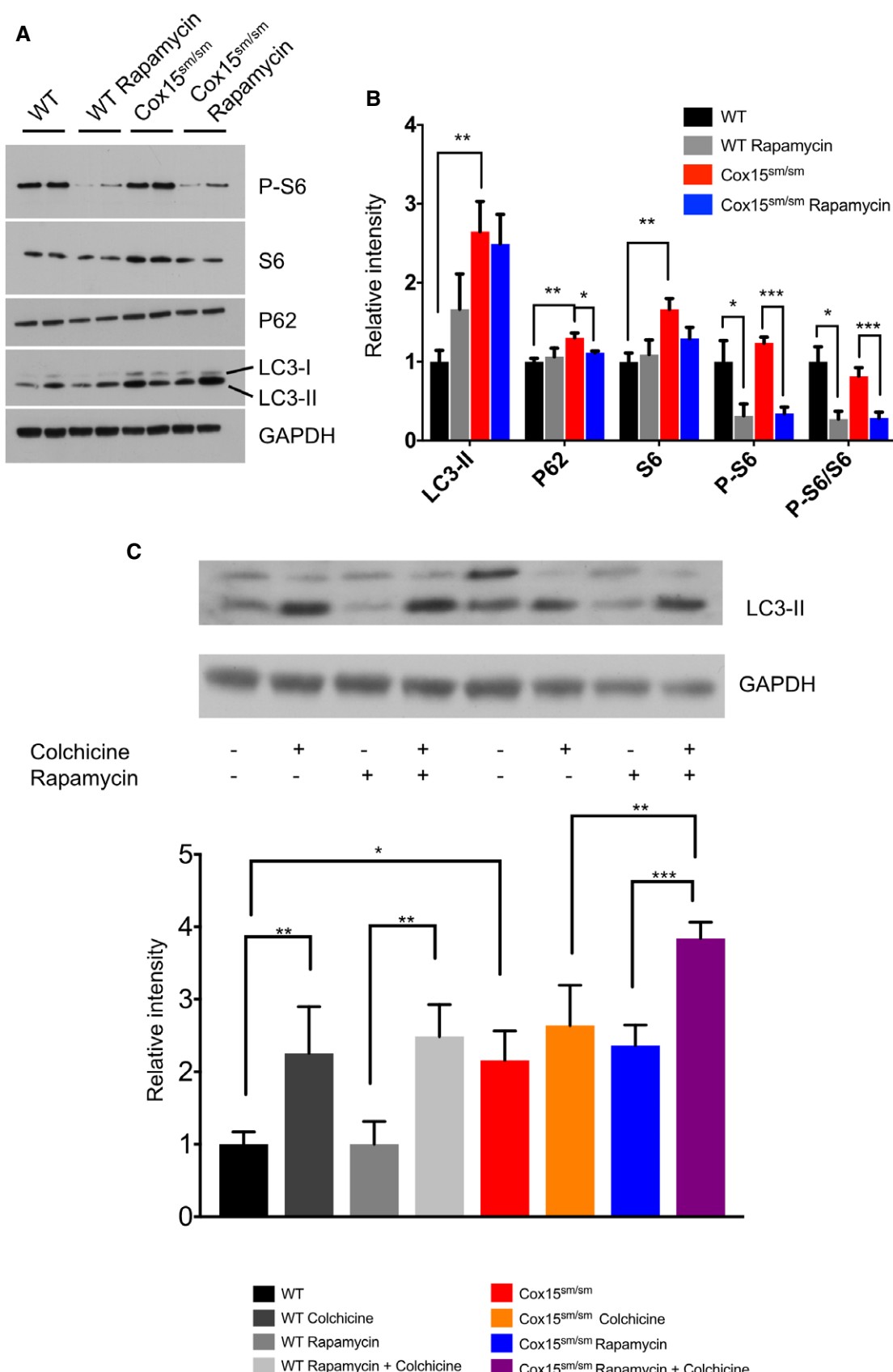

Figure 4.

◀

**Figure 4.  Rapamycin induces autophagy in *Cox15*^sm/sm^ muscles.**

A  Representative Western blot of autophagy markers at steady state in skeletal muscle. GAPDH was used as a loading control.

B  Densitometric analysis of immunoblots similar to that shown in (A) (*n* = 4/group). Black: untreated *WT*; gray: rapamycin-treated *WT*; red: untreated *Cox15*^sm/sm^; blue: rapamycin-treated *Cox15*^sm/sm^. Error bars represent SEM. The asterisks represent the significance levels calculated by one-way ANOVA with Tukey's post hoc multiple comparison test: **$P$ = 0.0068 (LC3-II WT versus *Cox15*^sm/sm^), **$P$ = 0.0063 (P62 WT versus *Cox15*^sm/sm^), *$P$ = 0.025 (P62 *Cox15*^sm/sm^ versus *Cox15*^sm/sm^+rapamycin), **$P$ = 0.0098 (S6 WT versus *Cox15*^sm/sm^), *$P$ = 0.034 (P-S6 WT versus WT+rapamycin), ***$P$ = 0.0002 (P-S6 *Cox15*^sm/sm^ versus *Cox15*^sm/sm^+rapamycin), *$P$ = 0.014 (P-S6/S6 WT versus WT+rapamycin), ***$P$ = 0.0007 (P-S6/S6 *Cox15*^sm/sm^ versus *Cox15*^sm/sm^+rapamycin).

C  Analysis of the autophagic flux (*n* = 3/group). Upper panel: representative Western blot of *Cox15*^sm/sm^ and WT muscles treated with colchicine, rapamycin, or rapamycin plus colchicine. Note that LC3-II was increased in *Cox15*^sm/sm^ versus *WT* samples. Colchicine did not increase LC3-II levels, suggesting a block in the autophagic flux; rapamycin plus colchicine-treated *Cox15*^sm/sm^ mice showed higher levels of LC3-II, suggesting that rapamycin increased the autophagic flux in *Cox15*^sm/sm^ muscles. Black: WT; dark gray: colchicine-treated WT; middle gray: rapamycin-treated WT; light gray: colchicine plus rapamycin-treated mice; red: untreated *Cox15*^sm/sm^; orange: colchicine-treated; blue: rapamycin-treated *Cox15*^sm/sm^; purple: colchicine plus rapamycin-treated *Cox15*^sm/sm^. Error bars represent SEM. The asterisks represent the significance levels calculated by one-way ANOVA with Tukey's post hoc multiple comparison test: **$P$ = 0.0049 (WT versus WT+colchicine), **$P$ = 0.0046 (WT+rapamycin versus WT+rapamycin+colchicine), *$P$ = 0.0225 (WT versus *Cox15*^sm/sm^), **$P$ = 0.0057 (*Cox15*^sm/sm^ colchicine versus *Cox15*^sm/sm^+rapamycin+colchicine), ***$P$ = 0.0003 (*Cox15*^sm/sm^ rapamycin versus *Cox15*^sm/sm^+rapamycin+colchicine).

selective elimination of dysfunctional mitochondria in *Cox15*^sm/sm^ muscles.

### Activation of autophagy via mTOR-dependent but not mTOR-independent pathway ameliorates the phenotype of *Cox15*^sm/sm^ mice

We then investigated autophagy in the skeletal muscle of rapamycin- and vehicle-treated naïve *Cox15*^sm/sm^ and *WT* littermates (Fig 4A and B). The phosphorylation of S6 (P-S6), a ribosomal protein which is a downstream target of mTORC1, was comparable in naïve *WT* versus *Cox15*^sm/sm^ muscles, but was clearly reduced in both genotypes upon rapamycin treatment, demonstrating that rapamycin was able to effectively inhibit mTORC1 *in vivo*. Total S6, which was increased in naïve *Cox15*^sm/sm^ versus *WT* muscles, returned to control levels by rapamycin treatment (Fig 4A and B). Since P-S6 levels reflect the protein translation rate, the reduced P-S6 levels we observed in rapamycin-treated animals suggest a reduction in protein translation, as expected in case of mTORC1 inhibition. P62, an autophagy substrate, was increased in naïve *Cox15*^sm/sm^ versus *WT* samples and normalized by rapamycin treatment (Fig 4A and B). Likewise, LC3-II, a marker of autophagosomes, accumulated in naïve *Cox15*^sm/sm^ samples and remained higher than in controls after rapamycin treatment (Fig 4A and B). Since the accumulation of LC3-II can be due to either an increase or a block of autophagy, we assessed autophagic flux *in vivo* by exposing both rapamycin-treated and naïve animals to colchicine, a blocker of lysosomal fusion with autophagosomes (Fig 4C and Appendix Fig S3). In basal conditions, *Cox15*^sm/sm^ muscles displayed significantly higher LC3-II levels than *WT*. Colchicine increased LC3-II in *WT* but not in *Cox15*^sm/sm^ muscles, suggesting a block of the autophagic flux in the latter. Rapamycin alone did not change LC3-II levels in either *WT* or *Cox15*^sm/sm^ samples. However, the combined exposure to rapamycin and colchicine significantly increased LC3-II levels in *Cox15*^sm/sm^ muscles, clearly indicating that rapamycin treatment was associated with an increase in the autophagic flux revealed by the addition of colchicine.

To investigate whether the mitophagy pathway could be involved in the clearance of dysfunctional mitochondria, we investigated the levels of PINK1 and Parkin in skeletal muscle-isolated mitochondria from rapamycin-treated and untreated mice of both genotypes. PINK1 clearly accumulated in mitochondria from *Cox15*^sm/sm^ versus *WT* mice (Appendix Fig S4), suggesting a reduced membrane potential in *Cox15*^sm/sm^ mitochondria, as we recently reported (Dogan *et al*, 2018). Parkin, which was undetectable in *WT* mitochondria irrespective of the treatment, was recruited to *Cox15*^sm/sm^ mitochondria (Appendix Fig S4). As observed for other mitochondrial proteins, PINK1 was significantly reduced in rapamycin-treated versus untreated *Cox15*^sm/sm^ muscle mitochondria (Appendix Fig S4), as a consequence of the increased autophagic flux. Mitochondrial Parkin showed a similar trend although the results displayed higher variability.

These results suggest that PINK1/Parkin-dependent mitophagy is activated in *Cox15*^sm/sm^ muscles, but is unable to efficiently clear damaged mitochondria. In contrast, rapamycin, by restoring the autophagic flux, significantly increased the elimination of dysfunctional mitochondria in the mutant mice.

To further investigate the role of autophagy in the rapamycin-mediated amelioration of the *Cox15*^sm/sm^ phenotype, we used rilmenidine, an agonist of imidazoline-1 receptor which regulates cAMP levels and acts as an mTOR-independent autophagy inducer.

Although rilmenidine increased the autophagic flux (Fig 5A), it had no effect on the motor performance of *Cox15*^sm/sm^ mice (Fig 5B), and failed to improve muscle morphology (Fig 5C–E).

Ultrastructurally, rilmenidine was ineffective to significantly reduce the number of damaged mitochondria (Appendix Figs S2 and S5) and led to a significant increase in mitochondria-containing autophagic vacuoles, compared to naïve and rapamycin-treated *Cox15*^sm/sm^ muscles (Appendix Fig S2). These results indicate that autophagy is not the only factor responsible for the rapamycin-mediated effect in *Cox15*^sm/sm^ mice.

### Rapamycin induces lysosomal biogenesis in *Cox15*^sm/sm^ skeletal muscle

Transcription factor EB (TFEB) is a master regulator of lysosomal biogenesis and autophagy. In its phosphorylated, inactive form, TFEB localizes to the cytosol, but upon dephosphorylation by calcineurin, it translocates to the nucleus where it increases the transcription of autophagy- and lysosome-related genes. Importantly, TFEB is a direct target of mTORC1, which inactivates it by phosphorylation. The link between TFEB and mTORC1 prompted us to

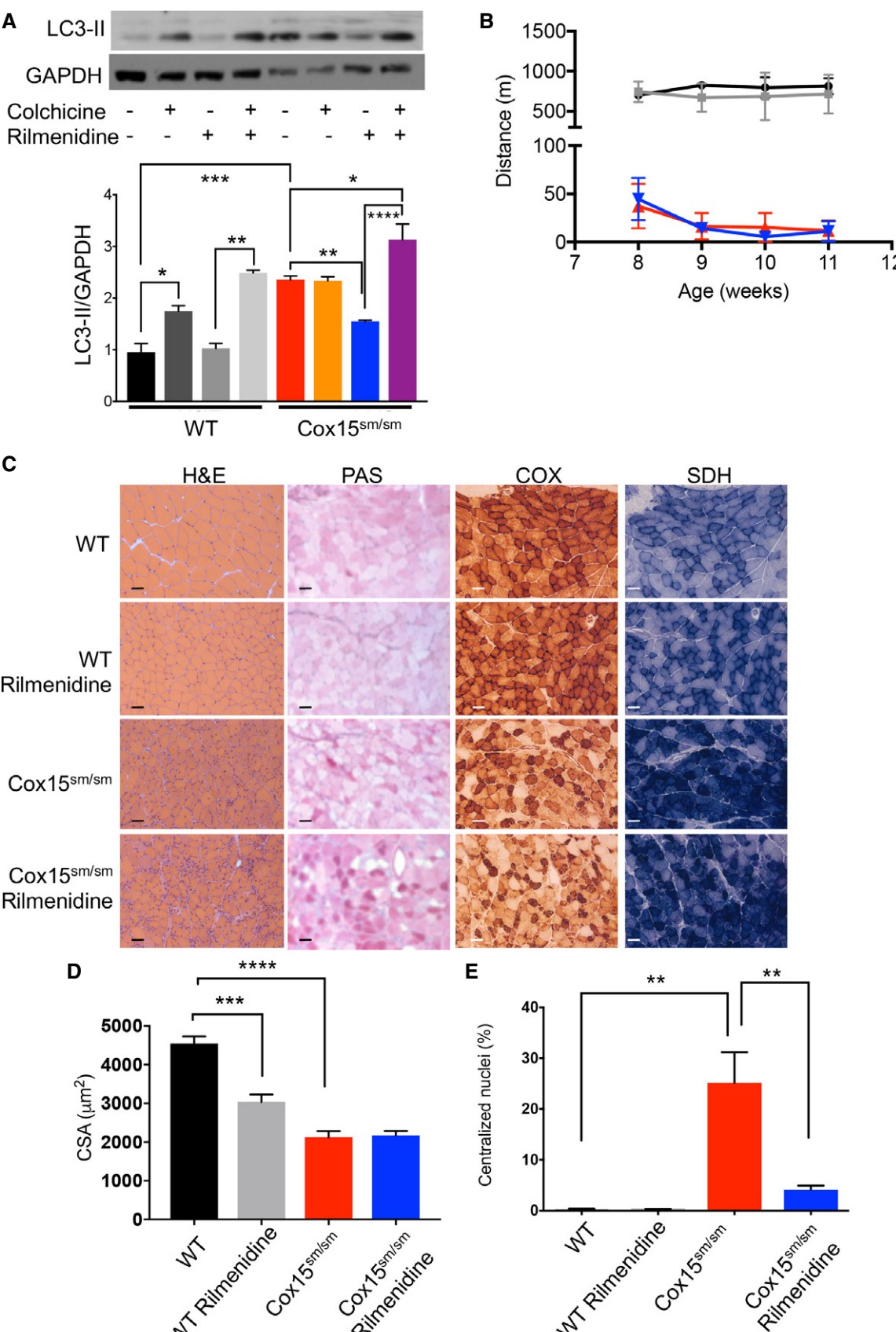

**Figure 5.**

**Figure 5.  Rilmenidine induces autophagic flux, but does not improve the phenotype of *Cox15sm/sm* mice.**

A    Analysis of the autophagic flux (*n* = 4/group). Upper panel: representative Western blot of *Cox15sm/sm* and *WT* muscles treated with colchicine, rilmenidine, or rilmenidine plus colchicine. Colchicine alone increased LC3-II in *WT* but not *Cox15sm/sm* mice; rilmenidine had no effect on WT LC3-II levels, but reduced them in *Cox15sm/sm* mice. Rilmenidine plus colchicine increased LC3-II levels in both *WT* and *Cox15sm/sm* mice. Black: WT; dark gray: colchicine-treated WT; middle gray: rapamycin-treated WT; light gray: colchicine plus rapamycin-treated mice; red: untreated *Cox15sm/sm*; orange: colchicine-treated; blue: rapamycin-treated *Cox15sm/sm*; purple: colchicine plus rapamycin-treated *Cox15sm/sm*. Error bars represent SEM. The asterisks represent the significance levels calculated by one-way ANOVA with Tukey's post hoc multiple comparison test: *$P$ = 0.012 (WT versus WT+colchicine), **$P$ = 0.0093 ((WT+rilmenidine versus WT+rilmenidine+colchicine), ***$P$ = 0.0001 (WT versus *Cox15sm/sm*), **$P$ = 0.0083 (*Cox15sm/sm* versus *Cox15sm/sm*+rilmenidine), *$P$ = 0.0112 (*Cox15sm/sm* versus *Cox15sm/sm*+rilmenidine+colchicine), **** $P$ < 0.0001 (*Cox15sm/sm*+rilmenidine versus *Cox15sm/sm*+rilmenidine+colchicine).

B    Treadmill analysis (*n* = 4/group). Black: untreated *WT*; gray: rilmenidine-treated *WT*; red: untreated *Cox15sm/sm*; blue: rilmenidine-treated *Cox15sm/sm*. Error bars represent SEM.

C    Histological and histochemical characterization of skeletal muscle in  rilmenidine-treated and untreated *Cox15sm/sm* and *WT* mice. H&E: hematoxylin and eosin; GT: Gomori trichrome; PAS: periodic acid–Schiff reaction. The scale bar corresponds to 50 μm.

D    Analysis of the cross-sectional area of muscle fibers in the different genotypes (*n* = 3/group). Black: untreated *WT*; gray: rilmenidine-treated *WT*; red: untreated *Cox15sm/sm*; blue: rilmenidine-treated *Cox15sm/sm*. Error bars represent SEM. The asterisks represent the significance levels calculated by one-way ANOVA with Tukey's post hoc multiple comparison test: ***$P$ = 0.0002, ****$P$ < 0.0001.

E    Analysis of the number of centralized nuclei in muscle fibers (*n* = 3/group). Black: untreated *WT*; gray: rilmenidine-treated *WT*; red: untreated *Cox15sm/sm*; blue: rilmenidine-treated *Cox15sm/sm*. Error bars represent SEM. The asterisks represent the significance levels calculated by one-way ANOVA with Tukey's post hoc multiple comparison test: **$P$ = 0.0091.

investigate TFEB localization upon rapamycin treatment. Immunofluorescence (IF) staining by an anti-TFEB antibody showed a significant increase in the number of TFEB-positive nuclei in rapamycin-treated *Cox15sm/sm* muscle compared to *WT* (Fig 6A and B). Notably, rilmenidine did not induce TFEB translocation to the nucleus.

In order to investigate whether increased nuclear localization of TFEB was accompanied by increased lysosomal biogenesis, we measured *LAMP1* transcript, LAMP1-positive vesicles, and cathepsin L activity in *Cox15sm/sm* and *WT* animals. Rapamycin-treated mice of both genotypes showed significantly increased *LAMP1* transcript levels (Fig 7A). IF with an anti-LAMP1 antibody (Fig 7B) showed a significantly higher number of lysosomes in naïve *Cox15sm/sm* compared to untreated and treated *WT* muscles (Fig 7B and C). Rapamycin further increased significantly the number of lysosomes in *Cox15sm/sm* samples (Fig 7B and C). Contrariwise, rilmenidine had no effect on LAMP1 staining (Fig 7B and C). No difference was detected in size among the LAMP1-positive vesicles (Appendix Fig S6). Finally, cathepsin L activity was significantly increased in rapamycin-treated compared to untreated *Cox15sm/sm* muscle samples (Fig 7D). These results suggest that sustained lysosomal biogenesis is fundamental to support efficient elimination of damaged mitochondria by autophagy.

## Discussion

A number of reports have recently provided evidence that the mTORC1 inhibitor rapamycin is able to partially ameliorate the phenotype of mitochondrial disease *in vivo*, e.g., mouse models and patient-derived mutant cells. For instance, administration of rapamycin by daily i.p. injections led to a remarkable prolongation of the lifespan, delay of the neurological derangement, and reduction of neuroinflammation in *Ndufs4−/−* mice, a model of severe complex I deficiency (Johnson *et al*, 2013). Post-onset oral administration of rapamycin at the same dose previously used i.p. (8 mg/kg) delayed the development of the encephalopathy in the *Ndufs4−/−* mouse (Felici *et al*, 2017). Also, neurons derived from induced pluripotent stem cells (iPSCs) from a patient with maternally

inherited Leigh syndrome (MILS), carrying a mutation in the MT-ATP6 gene, showed reduced glutamate toxicity after rapamycin treatment. This effect was accompanied by inhibition of protein synthesis and, as a consequence, sparing of ATP (Zheng *et al*, 2016). Reduced ATP consumption and proteotoxic stress and activation of autophagy were shown to contribute to the clinical and biochemical amelioration of several models of mitochondrial dysfunction, i.e., human cells, *C. elegans,* and mice (Peng *et al*, 2015). More recently, rapamycin was shown to inhibit the integrated mitochondrial stress response (ISRmt), a complex tissue-specific set of homeostatic pathways, involving transcriptional and metabolic adaptations as well as induction of the mitochondrial unfolded protein response (Khan *et al*, 2017). Finally, low-dose rapamycin was reported to prolong the lifespan of a thymidine kinase 2 (Tk2) knock-in mouse (Siegmund *et al*, 2017). Although this effect was not accompanied by modifications of the molecular hallmarks of the disease, RNAseq and metabolomics data suggested significant changes in amino acid, carbohydrate, and fatty acid metabolism in liver, with little effect on the brain.

Nevertheless, a general consensus on the mechanisms underlying the effects of rapamycin on primary mitochondrial dysfunction is still lacking. Notably, the contribution of autophagy was not thoroughly addressed in these works.

Here, we have shown that rapamycin-induced inhibition of mTORC1-dependent pathways plays a critical role in improving the clinical phenotype of a mouse model characterized by severe mitochondrial myopathy. While mTOR signaling was not significantly changed in naïve *Cox15sm/sm* versus *WT* mice, autophagy flux was reduced in *Cox15sm/sm* individuals, as demonstrated by increased levels of P62 and unchanged LC3-II levels by colchicine. This block was significantly overcome by the administration of either rapamycin or rilmenidine. However, the striking difference of the clinical and biochemical phenotypes by rapamycin versus rilmenidine suggests that autophagy induction is not sufficient to achieve the therapeutic effects and must be supported by the coordinated increase in lysosomal biogenesis to carry out the selective clearance of dysfunctional mitochondria. Although rapamycin seems to be a weak regulator of TFEB in cells (Settembre *et al*, 2012), the relatively prolonged treatment carried out in our experiments may have

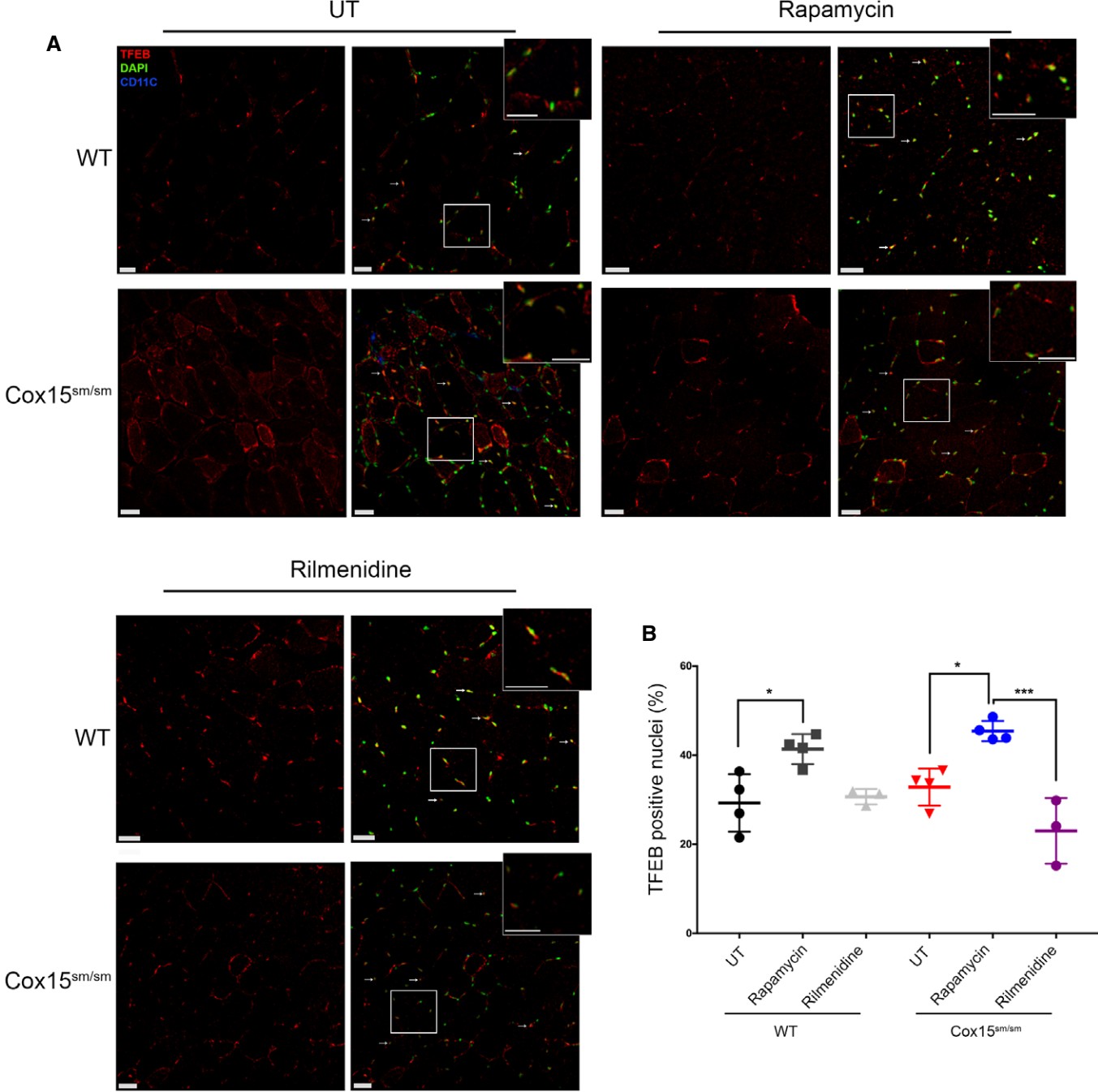

**Figure 6.  Rapamycin, but not rilmenidine, induces TFEB translocation to the nucleus.**

A    Anti-TFEB immunofluorescence on rapamycin- and rilmenidine-treated WT and *Cox15sm/sm* muscles versus untreated (UT) samples. CD11C (blue signal) indicates inflammatory cells. The scale bars correspond to 30 μm.

B    Quantification of the TFEB-positive nuclei (n = 3/group). Quantification was performed using Imaris spots surface excluding the CD11C-positive nuclei. Error bars represent SEM. The asterisks represent the significance levels calculated by one-way ANOVA with Tukey's post hoc multiple comparison test: *P = 0.0199 (WT versus WT+rapamycin), *P = 0.0150 (*Cox15sm/sm* versus *Cox15sm/sm*+rapamycin), ***P = 0.0001.

induced more robust TFEB activation. An additional possibility is that activation of transcription factor E3 (TFE3), another member of the MiT/TFE family of transcription factors, with partially overlapping activity with TFEB, may also contribute to increased lysosomal biogenesis under rapamycin treatment.

Rapamycin was more effective in improving the autophagic flux in *Cox15sm/sm* than in *WT* muscle. This is in agreement with recently reported data supporting a link between mitochondrial dysfunction and impairment of lysosomal activity (Demers-Lamarche *et al*, 2016). Several mechanisms have been proposed

**Figure 7.  Rapamycin, but not rilmenidine, increases lysosomal biogenesis.**

A   Real-time PCR analysis of *Lamp1* transcript (*n* = 3/group). Black: untreated *WT*; gray: rapamycin-treated *WT*; red: untreated *Cox15*<sup>sm/sm</sup>; blue: rapamycin-treated *Cox15*<sup>sm/sm</sup>. Error bars represent SEM. The asterisks represent the significance levels calculated by Student's *t*-test: **$P$ = 0.008, *$P$ = 0.038.

B   Anti-LAMP1 staining in rapamycin- and rilmenidine-treated *WT* and *Cox15*<sup>sm/sm</sup> muscles versus untreated (UT) samples. Rapamycin, but not rilmenidine, increases the number of LAMP1-positive vesicles in *Cox15*<sup>sm/sm</sup> samples (inset). The scale bars correspond to 30 μm.

C   Quantification of LAMP1-positive vesicles (*n* = 3/group). Error bars represent SEM. The asterisks represent the significance levels calculated by one-way ANOVA with Tukey's post hoc multiple comparison test: ****$P$ < 0.0001.

D   Cathepsin L specific activity (*n* = 3/group). Error bars represent SEM. The asterisks represent the significance levels calculated by Student's *t*-test: *$P$ = 0.0133.

to contribute to the reciprocal impact between these organelles. First, the molecular machineries involved in mitophagy, mitochondrial-derived vesicles (MDVs), mitochondrial-derived compartments (MDCs), and direct contact sites, mediate direct cross-communication between the two organelles. In particular, it has been recently shown that the lysosomal GTPase Rab7 controls not only lysosomal fission, but also mitochondrial fission via interaction with TBC1D15, which is targeted to mitochondria via interaction with Fis1, so that conditions affecting one organelle can also impact on the other (Wong *et al*, 2018). Second, mitochondrial dysfunction is a common finding in several lysosomal storage disorders, and, similarly, mitochondrial dysfunction may trigger lysosomal dysfunction (Plotegher & Duchen, 2017). Third, the master regulator of lysosomal

biogenesis TFEB has been shown to regulate also mitochondrial biogenesis and function (Mansueto *et al*, 2017).

A block of the autophagic flux may result in reduced clearance of dysfunctional mitochondria in *Cox15*<sup>sm/sm</sup> muscle. This hypothesis is supported by the accumulation of profoundly altered mitochondria in *Cox15*<sup>sm/sm</sup> samples. The increase in mitochondria in autophagic vesicles upon treatment with rilmenidine (Appendix Fig S2) confirms that mitochondria are delivered to lysosomes (as expected from the results on the autophagy flux experiment with colchicine), but they are not efficiently degraded, unlike what occurs with rapamycin.

Our results suggest that rapamycin induced the removal of damaged mitochondria, as demonstrated by correction of muscle morphology, reduction of mtDNA amount, normalization of CS

activity, and increased mitochondrial respiration. In addition, our data on PINK1/Parkin indicate that rapamycin activates a selective mitophagy pathway, which warrants future investigation.

In spite of the marked improvement in the muscle morphology and mitochondrial ultrastructure, the improvement in motor performance, albeit significant, is rather limited, and treated mice still perform much less than the *WT* littermates. Several reasons may explain this observation, including the relatively late age of the mice at the start of treatment (weaning), and the intrinsic limitations of our mouse model, which lacks an essential gene, so that the treatment can only have a limited impact on the biochemical defect. These effects were associated with both increased autophagic flux and lysosomal biogenesis. Contrariwise, rilmenidine treatment, which leads to activation of autophagy without the concurrent increase in lysosome biogenesis, completely failed to improve the clinical and morphological phenotype of $Cox15^{sm/sm}$ mouse muscles. Likewise, rilmenidine failed to ameliorate the disease progression in a SOD1 transgenic mouse model of amyotrophic lateral sclerosis (Perera *et al*, 2017).

In conclusion, we provide solid evidence that rapamycin induces substantial amelioration of mitochondrial function and ultrastructure, indicating efficient clearance of dysfunctional organelles by parallel activation of both autophagic flux and lysosomal biogenesis in skeletal muscle. Our results do not exclude that other mechanisms can be involved in mediating rapamycin effects. The reduction in phospho-S6 levels, clearly observed in the skeletal muscle of both treated *WT* and $Cox15^{sm/sm}$ animals, indicates that translation is indeed inhibited by rapamycin, and this may well contribute to the overall effect, as previously suggested (Peng *et al*, 2015; Zheng *et al*, 2016). Likewise, metabolic effects such as those reported by others (Johnson *et al*, 2013; Khan *et al*, 2017; Siegmund *et al*, 2017) are likely to have a role in the phenotypic amelioration of our mice. However, we propose that the activation of the TFEB-dependent transcriptional program, via rapamycin-dependent inhibition of the mTORC1 kinase activity, significantly contributes to the overall effect of rapamycin in ameliorating the phenotype of $Cox15^{sm/sm}$ mice.

Notably, rapamycin seems to be beneficial in all the models of mitochondrial disease so far analyzed, but its use in the treatment of mitochondrial patients raises several concerns, mainly related to side effects such as immunosuppression and glucose intolerance. However, preliminary evidence in healthy elderly volunteers showed that everolimus, a derivative of rapamycin, improved immune response to influenza vaccination in elderly people (Mannick *et al*, 2014). Overall, our data encourage additional studies in mice and humans to investigate the long-term effects and to test safety, tolerability, and eventually efficacy of rapamycin and rapalogs.

# Materials and Methods

### Reagents and materials

Antibodies anti-COX4, anti-UQCRC2, anti-NDUFA9, anti-ATP5A, anti-SDHA were from Abcam (OXPHOS cocktail ab110412, dilution 1:2,000); anti-SDHB (ab14714, dilution 1:200) and anti-COX1 (ab14705, dilution 1:2,000) were from Abcam; anti-HSC70 was from Santa Cruz (sc-7298, dilution 1:1,000); anti-GAPDH was from Abcam (ab8245, dilution 1:40,000); anti-P62 was from Abnova (H00008878-M01, dilution 1:1,000); anti-LC3-I/II was from Novus Biologicals (NB100-2220, dilution 1:1,000); anti-LAMP1 was from Cell Signaling (3243, dilution 1:1,000); anti-PINK1 was from Novus (BC100-494, dilution 1:1,000); and anti-Parkin was from Abcam (ab77924, dilution 1:500).

### Animal work

All procedures were conducted under the 1986 UK Animals (Scientific Procedures) Act and approved by Home Office license (PPL: 7538 and P6C97520A) and local ethical review. The mice were kept on a C57Bl6/129Sv mixed background, and wild-type littermates were used as controls. The animals were maintained in a temperature- and humidity-controlled animal care facility, with a 12-h light/dark cycle and free access to water and food, and were sacrificed by cervical dislocation.

### Behavioral analysis

Treadmill analysis (Panlab, Barcelona, Spain) was performed as described in Viscomi *et al* (2011). Briefly, motor exercise endurance was assessed according to the number of falls in the motivational air puff during a gradually accelerating program with speed initially at 6.5 m/min and increasing by 0.5 m/min every 3 min. The test was terminated by exhaustion, defined as > 10 falls/min into the motivational air puff.

### Oxygen consumption studies

Mouse muscles were homogenized in 0.075 M sucrose, 0.225 M mannitol, 1 mM EGTA, 0.01% fatty acid-free BSA (pH 7.4), and subtilisin 1 mg/g of muscle (Frezza *et al*, 2007). Mitochondria were isolated by differential centrifugation and resuspended in 25 mM sucrose, 75 mM sorbitol, 100 mM KCl, 0.05 mM EDTA, 5 mM $MgCl_2$, 10 mM Tris–HCl (pH 7.4), and 10 mM $H_3PO_4$, pH 7.4 (Fernandez-Vizarra *et al*, 2002).

For oxygraphic measurements, 75–150 μg of mitochondrial proteins was incubated in a buffer containing 225 mM sucrose, 75 mM mannitol, 10 mM Tris–HCl (pH 7.4), 10 mM KCl, 10 mM $KH_2PO_4$, 5 mM $MgCl_2$, and 1 mg/ml fatty acid-free BSA, pH 7.4. Oxygen consumption was evaluated by high-resolution respirometry using an Oroboros O2k apparatus (Oroboros Instruments, Austria), using the following substrate and inhibitor concentrations: 5 mM succinate and 2 μM rotenone for cII-dependent respiration; and 6 mM ascorbate and 300 μM TMPD and antimycin A 0.25 μg/ml for cIV-dependent respiration. 100 μM ADP was added to stimulate ATP-coupled oxygen consumption. 100 μM NaCN was added to completely inhibit respiration (Frezza *et al*, 2007).

### Blue native gel electrophoresis

For BNGE analysis, 250 μg of mitochondria isolated as described above was resuspended in native PAGE buffer (Invitrogen), protease inhibitors, and 4% digitonin and incubated for 1 h on ice before centrifuging at 20,000 *g* at 4°C. 5% Coomassie G250

**The paper explained**

**Problem**

Mitochondrial diseases are rare genetic disorders with extreme variability of symptoms, ranging from relatively benign myopathies to devastating encephalomyopathies. Mutations in either the nuclear or mitochondrial DNA can cause defects in the mitochondrial respiratory chain leading to disease. No cure is currently available for these conditions.

**Results**

We tested the efficacy of rapamycin in a mouse model of severe mitochondrial myopathy. Rapamycin is an inhibitor of MTORC1, a master regulator of several cellular pathways, including immune response, energy metabolism, and autophagy. We found that rapamycin markedly improves the structural and ultrastructural features of our mouse model by clearing non-functional mitochondria through the coordinated activation of autophagy and lysosomal biogenesis.

**Impact**

Rapamycin has been shown to be beneficial in a relatively large number of cellular and animal models of mitochondrial diseases. Our study suggests a new mechanism of action by rapamycin. These findings warrant future investigation to translate rapamycin or other inhibitors of MTORC1 into the clinical practice.

was added to the supernatant. 30 μg was separated by 3–12% gradient BNGE and either stained for in-gel activities or electroblotted on PVDF membranes for immunodetection (Nijtmans *et al*, 2002).

**Morphological analysis**

For histochemical analysis, tissues were frozen in liquid-nitrogen-pre-cooled isopentane. Eight-micrometer-thick sections were stained for COX and SDH, as described in Sciacco and Bonilla (1996). Analysis of cross-sectional area and centralized nuclei and analysis was performed on H&E-stained sections using ImageJ on four samples/genotype (600 fibers/sample). For ultrastructural studies, muscle samples were fixed in 2.5% glutaraldehyde in 0.1 M cacodylate buffer (pH 7.4) at 4°C, post-fixed in 2% $OsO_4$ in 0.1 M cacodylate buffer (pH 7.4) for 1 h, dehydrated in a graded series of ethanol and finally embedded in Epon resin. Thin sections were stained with lead citrate and uranyl acetate and evaluated with a transmission electron microscope (EM 109; Zeiss), as described in Cortese *et al* (2014).

For SDH, COX, and PAS analysis, five images per sample (six images for PAS) were acquired randomly in places of mixed-intensity fibers keeping the same light intensity and exposure settings using an Axio Observer Z1 with ApoTome 2 (Carl Zeiss Ltd.), composed of a Zeiss 10× ApoPlan objective and an AxioCam ICc1 camera. Images were acquired using Zen Pro software and analyzed with Fiji (Schindelin *et al*, 2012) (% of total area) keeping the threshold for all samples.

**Immunofluorescence and imaging**

For immunofluorescence, 8-μm frozen skeletal muscle sections were fixed in 4% PFA for 10 min, washed in PBS, and then incubated in 0.2% Triton in PBS for 15 min. After washing in PBS, sections were incubated in blocking solution (5% normal goat serum, 2% BSA, 1:40 M.O.M. blocking reagent in PBS) for 1 h at room temperature and then incubated with the primary antibodies diluted in DAKO Antibody Diluent with Background Reducing Components for 1 h at RT (1:1,000, anti-LAMP1, #ab24170 from Abcam; 1:200, anti-TFEB, #A303-673A, from Bethyl Laboratories; 1:200, Alexa Fluor 647 anti-mouse CD11c, #117312, from BioLegend). Slides were washed in PBS for 15 min and then incubated for 1 h at RT with Alexa Fluor secondary antibodies (1:300 in DAKO Antibody Diluent with Background Reducing Components). Sections were washed in PBS and mounted with ProLong Diamond Antifade Mountant with DAPI (Invitrogen). The images were acquired using a Dragonfly Spinning Disk imaging system (Andor Technologies Ltd.), composed of a Nikon Ti-E microscope, Nikon Apochromat 20×0.75 (for TFEB) and 100×1.49 TIRF objective, and an Andor iXon EMCCD camera. The Z-stacks were acquired using Fusion software (Andor Technologies) and the 3D images analyzed using Imaris software (Bitplane Inc.) creating spots surfaces for the whole image keeping the same conditions for all.

For SDH and PAS analysis, six images per sample were acquired randomly in places of mixed-intensity fibers keeping the same light intensity and exposure settings using an Axio Observer Z1 with ApoTome 2 (Carl Zeiss Ltd.), composed of a Zeiss 10× ApoPlan objective and an AxioCam ICc1 camera. Images were acquired using Zen Pro software and analyzed with Fiji (% of total area) keeping the same threshold for all samples.

**Biochemical analysis of MRC complexes**

Brain and skeletal muscle samples were snap-frozen in liquid nitrogen and homogenized in 10 mM phosphate buffer (pH 7.4). The spectrophotometric activity of cI, cII, cIII, and cIV, as well as citrate synthase (CS), was measured as described in Bugiani *et al* (2004).

**Real-time PCR**

RNA was isolated from skeletal muscles using Qiagen RNeasy Kit according to the manufacturer's instructions and retrotranscribed into cDNA using Omniscript Reverse Transcription Kit (Qiagen). Transcript analysis was carried out by SYBR Green Real-Time PCR, as described in Viscomi *et al* (2011). The primers 5′-CCTACGAG ACTGCGAATGGT-3′ and 5′-CCACAAGAACTGCCATTTTTC-3′ were used to amplify *Lamp1*.

**Western Blot analysis**

Mouse tissues were homogenized in ten volumes of 10 mM potassium phosphate buffer (pH 7.4). Mitochondrial-enriched fractions were collected after centrifugation at 800 *g* for 10 min in the presence of protease inhibitors, and frozen and thawed three times in liquid nitrogen. Protein concentration was determined by the Bradford protein assay (BioRad, Watford, UK). Aliquots, 70 μg each, were run through a 4–12% SDS–PAGE and electroblotted onto a nitrocellulose membrane, which was then immunodecorated with different antibodies.

**Statistical analysis**

All numerical data are expressed as mean ± standard error of the mean (SEM). One- or two-way ANOVA with Tukey's post hoc test was used for multiple comparisons. Unpaired Student's *t*-test was used for pairwise comparison of independent experimental groups (see legends). Differences were considered statistically significant for $P < 0.05$. Exact *P*-values and number of animals ($n$) for each experiment are in the legends. Power analysis was conducted assuming a significance level of 0.05 and a probability of detecting the effect of 80%. Animals were randomized to the different groups based on the appropriate genotype. No animals were excluded from analysis. No blinding procedure was used because of the obvious clinical and morphological phenotypes of the animals the study. We assumed normal distribution. No particular method was used to determine whether the data met assumptions of the statistical approach.

**Expanded View** for this article is available online.

## Acknowledgements

This work was supported by a core grant of the MRC to the Mitochondrial Biology Unit (MC_UU_00015/5), the ERC grant FP7-322424 (to MZ), and the NRJ-Institut de France Grant (to MZ). We thank David C. Rubinsztein for useful discussion and for providing tools and reagents. We are grateful to the personnel at Phenomics and ARES animal care facilities for skillful technical assistance.

## Author contributions

GC and SAD carried out the in vivo phenotype analysis and performed biochemical measurements and molecular analysis; CB performed the microscopy analysis; MM, CL and GF performed the EM experiments; RC carried out histological, histochemical, and morphometric analyses; GC, SAD, RC, and CV analyzed the data; and GC, CV, and MZ conceived the study and wrote the manuscript .

## Conflict of interest

The authors declare that they have no conflict of interest.

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
