## [Review Process File · EMBO Molecular Medicine]

Rapamycin rescues mitochondrial myopathy via coordinated activation of autophagy and lysosomal biogenesis

Gabriele Civiletto, Sukru Anil Dogan, Raffaele Cerutti, Gigliola Fagiolari, Maurizio Moggio, Costanza Lamperti, Cristiane Benincà, Carlo Viscomi and Massimo Zeviani

Review timeline:

Submission date:	19 December 2017
Editorial Decision:	28 February 2018
Revision received:	05 July 2018
Editorial Decision:	20 August 2018
Revision received:	12 September 2018
Accepted:	17 September 2018

Editor: Céline Carret

Transaction Report:

1st Editorial Decision

28 February 2018

Thank you for the submission of your manuscript to EMBO Molecular Medicine. We have now heard back from the three referees whom we asked to evaluate your manuscript.

As you will see from the reports below, the referees find the topic of your study of potential interest. However, they also raise a number of serious concerns about the conclusiveness of the results and quite a few technical issues. Referee 1 feels that the study is too preliminary at this stage and more experimental data are needed to validate the results. Referee 2 finds the paper descriptive and a new version should include data performance, analysis, and most importantly, interpretation. Referee 3 also found overstatements and over-interpretations of the findings and would like to see additional data and repeated experiments to make the conclusions stronger and the data more meaningful.

Overall it is clear that publication of the manuscript cannot be considered at this stage. I also note that addressing the reviewers concerns in full will be necessary for further considering the manuscript in our journal and this appears to require a lot of additional work and experimentation. I am unsure whether you will be able or willing to address those and return a revised manuscript within the 3-6 months deadline. On the other hand, given the potential interest of the findings, I would be willing to consider a revised manuscript with the understanding that the referees' concerns must be fully addressed and that acceptance of the manuscript would entail a second round of review. I would add that it is particularly important that all of their suggestions [note that this would include the additional experiments, more mechanism, rephrasing and tuning down the claims] are taken on board as we cannot consider its publication otherwise.

I should remind you that EMBO Molecular Medicine encourages a single round of major revision only and that, therefore, acceptance or rejection of the manuscript will depend on the completeness of your responses included in the next, final version of the manuscript. For this reason, and to save you from any frustrations in the end I would strongly advise against returning an incomplete revision and would also understand your decision if you chose to rather seek rapid publication elsewhere at this stage.

Should you decide to revise your article for EMBO Molecular Medicine, revised manuscripts should be submitted within three months of a request for revision; they will otherwise be treated as new submissions, except under exceptional circumstances in which a short extension is obtained from the editor.

Should you find that the requested revisions are not feasible within the constraints outlined here and choose, therefore, to submit your paper elsewhere, we would welcome a message to this effect.

I look forward to receiving your revised manuscript.

***** Reviewer's comments *****

Referee #1 (Remarks for Author):

The study by Civiletto and coauthors provides evidence that Rapamycin administration for a period of four weeks induces phenotype amelioration in a skeletal-muscle-restricted mitochondrial disease model. These effects are substantially caused by the activation of autophagy and the increase of TFEB-related lysosomal biogenesis.

This work is in principle of interest but, however, whilst the results presented are generally promising, the data quality is not good enough: at present, in most cases the shown Western blots appear to be oversaturated (see for instance Fig 2B, Fig 4A, Fig 4C and Fig 5A). Further, the study is still preliminary at this point. There are some issues summarized here below, which should be promptly addressed to largely improve the manuscript:

- How do the authors explain the reduction of mitochondrial proliferation with increase of CS activity, higher levels of respiratory chains subunits and mitochondrial DNA in Cox15-sm/smmice? Moreover, the fact that LC3 II and P62 levels are higher in Cox15-sm/sm, as also confirmed by Colchicine experiment (the comparison between wt vs Cox15-sm/sm mice upon colchicine treatment is needed), suggests that in these mice there is an alteration of lysosome efficiency. How do the authors explain this possible defect?

- Authors should address whether or not mitophagy is activated upon Rapamycin treatment: i.e. as for the PINK1-PARKIN cascade activation.

- TFEB activation is a key point of this manuscript; thus, it should be better analyzed by performing Real Time PCR of some TFEB substrates, and by evaluating CTSD protein levels. In Figure 6 B, the authors should quantify the size of LAMP1 positive structures in basal levels from Cox15-sm/sm mice (it looks like there are enlarged LAMP1 structures in these mice): indeed, a dimensional analysis could reveal the presence of abnormally-enlarged structures (autophagolysosomes), presumably with accumulated substrates, a clear feature of lysosome defects. EM analysis could help out to clarify this point. Also, I suggest to show enlarged images of LAMP1 staining.

- The number of LAMP1 structures is higher only in Cox15-sm/sm mice, but not in WT mice,

while TFEB activation is present in both mouse models; how do they explain this controversial point?

- Could Rapamycin treatment promote also the activation of TFE3 (Martina et al. 2016) or other autophagy transcription factors, mediated by inhibition of mTORC1 (Saxton and Sabatini, 2017), that could explain the beneficial effects described in Cox15-sm/sm mice?

Referee #2 (Remarks for Author):

This is an overall well-written, thoughtful, and interesting paper that interrogates the mechanism of autophagy inhibition strategies in a COX10 deficient model of mitochondrial myopathy. Two autophagy inhibition strategies are compared that are either mTOR dependent (rapamycin) or mTORC independent (rilmenidine, acts by modulation cAMP levels). However, the current data, which are marginal to modest in magnitude and more qualitative in description than quantitative in degree for most parameters measured, appear to be overinterpreted by authors and do not currently substantiate the authors claim that "these data indicate that rapamycin treatment remarkably ameliorates the myopathic phenotype". Further they do not show any evidence for their statement at the end of the results section that any improvements observed from rapamycin treatment result "possibly through selective elimination of dysfunctional mitochondria in Cox15sm/sm muscles." Autophagic flux experiments are well-done and compelling, but the conclusions drawn appear to be overstated and not fully substantiated by the data shown. The final conclusion that these data support clinical use of rapamycin in human patients with mitochondrial disease are not substantiated and need to be tempered given potential adverse effects (glycogen storage) shown here and no prior clinical trials in this human disease population.

Major Concerns:

1. Results 1st section detailing rapamycin effects in the COX10 model are largely descriptive in text, with no magnitude of effect discussed for any of the histochemical dyes used to draw conclusions

2. Results discuss that "PAS staining also revealed an accumulation of glycogen in skeletal muscle of rapamycin-treated but not in naïve Cox15sm/sm mice." This seems highly concerning, not beneficial, since glycogen storage in muscle is itself a cause of a class of muscle disease. This should be more appropriately discussed, including possible negative implications. A quantitative measure of glycogen enrichment should be provided to assess magnitude of effect beyond one section shown in Fig 1B. It also seems concerning that rapamycin treatment in wild-mice significantly reduces muscle size (CSA, Fig 1C) by 25% -- this is not mentioned in text or discussed, but needs to be carefully included for possible toxicity or rapamycin treatment.

Similarly, results state "COX and COX/SDH histochemistry in skeletal muscle showed increased number of COX-positive fibers, and a parallel reduction of SDH hyperintense fibers, another index of mitochondrial proliferation, in treated vs. naïve Cox15sm/sm animals". However Fig 1B shows persistence of substantial abnormalities in COX and COX/SDH in rapamycin-treated mice. The magnitude of the effect should be quantified and determined if really significantly different from untreated mice in multiple measurements, not just in single image shown.

3. The improved motor performance on treadmill test is described as "remarkable", when the Fig 1A shows a significant but very marginal magnitude of effect, where relative to normal baseline (700 m) the mutant mice go from 150-200 m up to 200-250 m at different time points. This is not really remarkable.

4. Fig 2A data shows a significant but not complete resolution regarding CS or CIV/CS activities, making any change partial but not "returned to normal values" as stated in results

section

5. Results and Fig 2B legend states "Protein levels of respiratory chain subunits were reduced in rapamycin-treated vs. naïve Cox15sm/sm mice, to levels comparable to WT (Figure 2B)". However, in the immunoblot shown only UQCRC2 and NDUFA9 levels seem to be less in treated than untreated COX10 mice, and no consistent change with the other 3 complex subunits tested (ATP5A, SDHA, COX4 - where COXIV antibody did not seem to work well in any condition). These data should be replicated and statistical analyses performed on a quantified (ImageJ or otherwise) analysis of replicate data. Similarly, the BNG data shown in Fig 1C are not clearly showing a substantial increase in the band marked "cIV", and these need to be replicated and quantified relative to a loading control; in Fig 1D normalized quantitative data from multiple replicate experiments needs to be shown as well to evaluate the magnitude of the effect, which seems marginal at best relative to levels in WT controls.

6. Fig 3 conjectures that mitochondria alterations "possibly result from partially digested organelles within endolysosomes"... but no evidence of this is shown.

7. The autophagy characterization and flux analyses with in vivo colchicines exposure are well-done and compelling. However, the results conclusion that "These results demonstrate that rapamycin increased the autophagic flux in Cox15sm/sm muscles, and suggest this as a mechanism operating the elimination of dysfunctional mitochondria in the mutant mice" are not substantiated, as at no time do the authors demonstrate "elimination of dysfunctional mitochondria in the mutant mice".

8. Based on the rilmenidine experiments, the authors conclude "autophagy is not the only factor responsible for the rapamycin-mediated effect in Cox15sm/sm mice". As mTORC1 is known to regulate both autophagy and translation, the investigators should also interrogate translation in their animals, which has been shown previously to contribute together with autophagy inhibition to rapamycin effect in mitochondrial disease model animals and cells (PMID: 26041819). However, this citation is not included in the manuscript and no consideration is given to this major mechanistic aspect of mTORC1 inhibition as a therapeutic strategy in mitochondrial disease.

These experimental data also appear to be misstated in the conclusion, "Rilmenidine failed to improve the clinical and morphological phenotype, and in fact caused further reduction of fiber size and worsening of the dystrophic lesions of Cox15sm/sm mouse muscles". However, this is not what the data shown in Figs 5C (immunohistochemistry) or 5D (CSA).

9. The results in the 3rd section indicate COX10 mutant mice already have increased LAMP1 and TFEB expression, and rapamycin only exacerbates this but does not normalize it at all toward wildtype. In contrast, rilmenidine DOES normalize TFEB expression. Thus, it is not clear how this supports the authors' conclusion that, "These results suggest that sustained lysosomal biogenesis is fundamental to support efficient elimination of dysfunctional mitochondria by autophagy." Isn't this partly obvious, however, since lysosomes are an essential part of the autophagy process, not distinct from it? Further, as above, there is no data showing individual mitochondrial function level and clearance by autophagy to substantiate this claim. Finally, additional experiments with TFEB knockouts would be needed to support the Conclusion the authors make that "We propose that this effect may be mediated, at least in part, by activation of the Tfeb-dependent transcriptional programme:

10. The final statement in the conclusion is irresponsible and very concerning: "Overall, our data encourage the use of rapamycin or its derivatives in the treatment of mitochondrial diseases." No clinical trial has been performed in human subjects with mitochondrial disease, and this statement should be tempered appropriately to recommend clinical trials to assess efficacy, tolerability and safety (given some of the concerning findings shown in this work), NOT clinical use.

Minor Concerns:

1. The references are outdated to describe mitochondrial disease (2004), and tend to self-cite the authors as opposed to giving more updated and comprehensive references to this field that has changed substantially since 2004
2. Results section 1 reference growth restriction in Supp Fig S1, but this supplemental file shows mouse ultrastructure only.
3. For the rilmenidine-treatment histochemical analyses shown in Fig 5C, no quantitation is given. The conclusion is that there is no therapeutic effect, but COX/SDH changes shown in Fig 5C with Rilmenidine treatment seem highly similar to what was shown in Fig 1B with rapamycin treatment, except the conclusion in the latter only was that a therapeutic effect was seen. Both should be plotted together on one graph relative to wild-type to fully assess whether any histologic response occurred in either treatment model.
4. Fig 6A - It is not clear what the arrows are pointing too re: "brown staining", as only a portion of the brown staining within nuclei appears to be labeled by white arrows in each panel. Co-staining with a nuclear dye would be helpful to evaluate their statement that rilmenidine did not cause Tfeb translocation to the nucleus.
5. A few spelling and grammatical errors occur throughout, which should be corrected (eg, 'Contrariwise' is not really a word; a "significantly increase" should be "significant increase", gene names should be italicized and not stated as "gene" after the gene name, etc).

Referee #3 (Remarks for Author):

The manuscript investigates the effect of long term rapamycin treatment in a mouse model of mitochondrial disease.

The immunohistochemistry presented in Fig 1B is generally convincing, as is the EM imaging. Other sections of the results are not quite so clear cut.

Sm/sm +Rap shows less SDH staining than sm/sm -Rap, however the cells do not look especially COX positive, as the DAB staining is weak, and the quantitation is not evaluating the change in COX positive fibres, which would be more convincing.

P8- The lack of change in CIV activity or indeed relative increase shown in Fig 2A does not quite correspond with the decrease in COX4 by western blot in panel B. Similarly, there is a mild increase in CIV activity in BN gels both in wild type and Cox15sm/sm after rapamycin treatment that does not quite agree with the data in 2B, where COXIV is up in WT +R and down in sm/sm +R. Thus, the text "No differences were detected between treated vs. naïve WT animals" is not a completely accurate reflection of the data as WT+R does show differences in COXIV.

I would suggest that the word 'slightly' or some equivalent be added to "immunovisualization confirmed an 'slightly' increase in COX activity and amount of fully assembled cIV in rapamycin-treated vs. naïve Cox15sm/sm muscle samples (Figure 2C,D)" this would also be more consistent with the phrasing used in the legend.

For consistency Fig 3 should have arrows to indicate mitochondria in the WT panel. The size bar is only visible in one panel, for consistency it would be better to be present in all. This is true for other figures.

P9 - the text state "In basal conditions Cox15sm/sm muscles displayed significantly higher LC3-II levels than WT." The significance is not shown

on the graphical representation below so it would be more accurate to show the statistical significance or change the wording.

The text states that "Colchicine increased LC3-II in WT but not in Cox15sm/sm muscles,". The increase in WT is very robust but a distinct change also occurs in the sm/sm sample (lane 5 of lane 6). The levels of LC3-II in both lanes 6 and 8 are higher than lanes 5 and 7. The text describing these changes upon treatment is not entirely consistent with the images presented, which are from single examples of each condition. With mouse expts it is not uncommon to show that the effect is consistent by showing a number of samples from different animals, as with fig2 and panel A in the same figure. The description of the result would be more convincing if it were seen in multiple samples.

Since the colchicine alone as well as in combination with rapamycin increased LC3-II levels in sm/sm mice, this reviewer is not convinced by the conclusion that "These results demonstrate that rapamycin increased the autophagic flux in Cox15sm/sm muscles,".

Fig 5C has no size bars on the IHC. The fibre size seems to change with +Ril treatment but it is not possible to evaluate this without size bars. The sm/sm +Ril seems to change to have a more varied fibre size. WT H&E in particular looks very different in agreement with the representation in panel D, but all the other WT sections look to have a much smaller CSA than the WT H&E stained section. This does not look consistent across the panel and would be expected to give a larger error bar than seen in panel D for the WT across antibody staining panels.

The legend for Fig 6A states "Note that the increase of the brown staining in the nuclei of rapamycin- but not rilmenidine-treated muscles. Right panel: quantification of n=3 animals/group." This is not evident even when the image is magnified many fold. The data is transformed into a plot to the right of the sections but does not appear to reflect what can be seen in the TFEB staining either in WT or sm/sm samples.

Similarly, the Anti-Lamp1 staining and the quantification are not convincing. Is the quantification looking at number or intensity of signal. All the WT signals are weaker but the decrease in number of foci with rilmenidine looks to be the same in both WT and sm/sm.

The conclusion is that rapamycin could be used for the treatment of all mitochondrial diseases, however, the discussion also stated that rapamycin treatment was "accompanied by inhibition of protein synthesis...", which does not seem like a positive outcome, as it would also induce mitochondrial dysfunction.

The legends frequently repeat the main text and give an interpretation of the result rather than just an explanation of what is being presented. This is not EMM style and is repetitive and in some instance not consistent with the main text.

Minor points

Some methods are brief. Antibodies should give catalogue numbers.

P4 - 'protein translation' - should read 'protein synthesis'.

'skeletal muscle' is written both with and without a hyphen, these should probably be 2 separate words, but should be consistent throughout.

P13 - muscle spelling needs correcting.

P19 - "into cDNA using XXX kit" the details of the kit used are absent.

Some of the phrasing is not very appropriate eg 'actually' 3 lines from the end of the discussion

***** Reviewer's comments *****

Referee #1 (Remarks for Author):

The study by Civileto and coauthors provides evidence that Rapamycin administration for a period of four weeks induces phenotype amelioration in a skeletal-muscle-restricted mitochondrial disease model. These effects are substantially caused by the activation of autophagy and the increase of TFEB-related lysosomal biogenesis.

This work is in principle of interest but, however, whilst the results presented are generally promising, the data quality is not good enough: at present, in most cases the shown Western blots appear to be oversaturated (see for instance Fig 2B, Fig 4A, Fig 4C and Fig 5A).

We repeated and quantified all the blots requested by the reviewer.

Further, the study is still preliminary at this point. There are some issues summarized here below, which should be promptly addressed to largely improve the manuscript:

- How do the authors explain the reduction of mitochondrial proliferation with increase of CS activity, higher levels of respiratory chains subunits and mitochondrial DNA in Cox15-sm/smmice?

I suspect we did not make this point very clear. CS and mtDNA copy number are indexes of mitochondrial mass. They are increased in KO mice compared to controls, as a “compensatory” but ineffective consequence of the respiratory chain defect, but they are significantly *reduced* in rapamycin-treated KO animals to WT values. The rapamycin-KO mice show *reduced* mitochondrial content but increased activity of the respiratory chain complexes, an effect which we explain by a clearing of dysfunctional mitochondria. Those mitos that survive work better, as suggested by increased respiration.

Moreover, the fact that LC3 II and P62 levels are higher in Cox15-sm/sm, as also confirmed by Colchicine experiment (the comparison between wt vs Cox15-sm/sm mice upon colchicine treatment is needed), suggests that in these mice there is an alteration of lysosome efficiency. How do the authors explain this possible defect?

We agree with the reviewer that our data suggest an impairment of lysosomal degradation. Accumulating evidence in the literature support the idea that lysosomes and mitochondria reciprocally impact each other, and several mechanisms have been proposed to contribute. First, the molecular machineries involved in mitophagy, mitochondrial derived vesicles (MDVs), mitochondrial derived compartments (MDCs) and direct contact sites mediate direct cross-communication between the organelles (reviewed in Audano et al, J Neurochemistry, 2018). In particular, it has been recently shown that the lysosomal GTPase Rab7, not only controls lysosomal fission, but also mitochondrial fission via interaction with TBC1D15, which is targeted to mitochondrial via interaction with Fis1, so that conditions affecting one organelle can also impact on the other (Wong, Nature 554; 382-385, 2018). Second, mitochondrial dysfunction is a common finding in several lysosomal storage disorders, and, similarly, mitochondrial dysfunction may trigger lysosomal dysfunction (Plotegher & Duchon 2017).

Third, the master regulator of lysosomal biogenesis TFEB has been shown to regulate also mitochondrial biogenesis and function (Mansueto et al, Cell Metab, 2016). Interestingly, it has been shown that the PARKIN Q311X mutation alters mitochondrial quality control via downregulation of both PGC-1 α and TFEB via activation of the transcriptional repressor PARIS (Siddiqui et al. 2015). This effect was reversed by administration of rapamycin in a TFEB-dependent manner.

We expanded the discussion according to the above considerations.

- Authors should address whether or not mitophagy is activated upon Rapamycin treatment: i.e. as for the PINK1-PARKIN cascade activation.

We performed analysis of Pink1/Parkin pathway. However, the poor quality of the antibodies in vivo does not allow to give interpretable results: please refer to a representative immunoblot obtained by using the anti-Parkin antibody. The Pink1 antibody gave a virtually blank film.

As the reviewer can appreciate, multiple bands are present at a suitable MW, making it impossible to draw any meaningful conclusion. This probably why most of the studies on the characterization of Pink1 and Parkin found in the literature are done in cell cultures and are based on the use of tagged proteins.

- TFEB activation is a key point of this manuscript; thus, it should be better analyzed by performing Real Time PCR of some TFEB substrates, and by evaluating CTSD protein levels.

We analysed Lamp1 as a prototypical direct target of TFEB. Both Lamp1 protein and transcript were significantly upregulated in rapamycin-treated samples. The transcripts and protein levels of Cathepsins did not change, but the activity of Cathepsin L was significantly higher in rapamycin-treated muscles. It is important to stress that most the data present in the literature are based on short-term treatments in cell cultures, and it is well possible that long-term treatments in vivo may lead to results different from acute treatments.

In Figure 6 B, the authors should quantify the size of LAMP1 positive structures in basal levels from Cox15-sm/sm mice (it looks like there are enlarged LAMP1 structures in these mice): indeed, a dimensional analysis could reveal the presence of abnormally-enlarged structures (autophagolysosomes), presumably with accumulated substrates, a clear feature of lysosome defects. EM analysis could help out to clarify this point. Also, I suggest to show enlarged images of LAMP1 staining.

We analysed the size of small (<math><0.5\mu\text{m}</math>), medium (<math>0.5 <x < 1\mu\text{m}</math>) and (>math>1\mu\text{m}</math>) LAMP1-positive vesicles but no differences were observed. These data are now included in figure EVXX.

- The number of LAMP1 structures is higher only in Cox15-sm/sm mice, but not in WT mice, while TFEB activation is present in both mouse models; how do they explain this controversial point?

LAMP1 positive vesicles are already higher in untreated KO vs WT, probably not because of increased TFEB/TFE-mediated biogenesis but because of reduced degradation, as suggested by the fact that LAMP1 transcript was similar in the two genotypes. Rapamycin instead induces lysosomal biogenesis as demonstrated by increased LAMP1 transcript.

- Could Rapamycin treatment promote also the activation of TFE3 (Martina et al. 2016) or other autophagy transcription factors, mediated by inhibition of mTORC1 (Saxton and Sabatini, 2017), that could explain the beneficial effects described in Cox15-sm/sm mice?

As the referee said, TFEB and TFE3 are partly redundant, and we agree that it is likely that both mediate the effect of rapamycin.

Referee #2 (Remarks for Author):

This is an overall well-written, thoughtful, and interesting paper that interrogates the mechanism of autophagy inhibition strategies in a COX10 deficient model of mitochondrial myopathy. Two autophagy inhibition strategies are compared that are either mTOR dependent (rapamycin) or mTORC independent (rilmenidine, acts by modulation cAMP levels). However, the current data, which are marginal to modest in magnitude and more qualitative in description than quantitative in degree for most parameters measured, appear to be overinterpreted by authors and do not currently substantiate the authors claim that "these data indicate that rapamycin treatment remarkably ameliorates the myopathic phenotype".

We would like to point out that doubling (from 83 to 172 m) the motor performance of our Cox15 KO model, lacking an essential enzyme of hemeA biosynthesis, is not a trivial result. In addition, we want to stress that the most remarkable effect of rapamycin is the improvement in muscle morphology. However, we agree with the reviewer that there were some overstatements that we now converted into more realistic conclusions.

Further they do not show any evidence for their statement at the end of the results section that any improvements observed from rapamycin treatment result "possibly through selective elimination of dysfunctional mitochondria in Cox15^{sm/sm} muscles."

We quantified the number of disrupted mitochondria in EM specimens, showing a significant reduced upon rapamycin-treatment compared to untreated naïve KO muscles. Together with the results on autophagic flux, these data support the idea that rapamycin induces the clearance of dysfunctional mitochondria by reactivating autophagy.

Autophagic flux experiments are well-done and compelling, but the conclusions drawn appear to be overstated and not fully substantiated by the data shown. The final conclusion that these data support clinical use of rapamycin in human patients with mitochondrial disease are not substantiated and need to be tempered given potential adverse effects (glycogen storage) shown here and no prior clinical trials in this human disease population.

What we meant is that our results support the beneficial effects of rapamycin in mitochondrial myopathies and encourage further investigations. This may well include the use of rapamycin (or rapalogs) in human studies, to evaluate its potential application in the clinics. We rephrased this in the new version.

Major Concerns:

1. Results 1st section detailing rapamycin effects in the COX10 model are largely descriptive in text, with no magnitude of effect discussed for any of the histochemical dyes used to draw conclusions

In the original manuscript, we only quantified CSA and centralized nuclei on H&E. Histochemical assays are in principle qualitative, and we have used spectrophotometric assays to quantitatively support the histoenzymatic results. However, we have now quantified SDH staining and glycogen content; these results are now included in figure 1

2. Results discuss that "PAS staining also revealed an accumulation of glycogen in skeletal muscle of rapamycin-treated but not in naïve Cox15^{sm/sm} mice." This seems highly concerning, not beneficial, since glycogen storage in muscle is itself a cause of a class of muscle disease. This should be more appropriately discussed, including possible negative implications. A quantitative measure of glycogen enrichment should be provided to assess magnitude of effect beyond one section shown in Fig 1B. It also seems concerning that rapamycin treatment in wild-mice significantly reduces muscle size (CSA, Fig 1C) by 25% -- this is not mentioned in text or discussed, but needs to be carefully included for possible toxicity or rapamycin treatment.

As mentioned above, glycogen content has now been quantified. By EM examination glycogen in muscle fibers is in the cytosol, not in lysosomes (as in Glycogenosis II). The amount of glycogen in cytoplasm is pathological when there is evidence that glycogen cannot be utilized to supply phosphorylated glucose, but an increase of glycogen per se can be caused by non-pathological conditions, for instance by an increase in the glucose transporter Glut4, as it has been demonstrated

in TFEB-treated animals (Mansueto et al 2017). We do not have any evidence that the accumulation of glycogen is due to impaired release of glucose or altered structure of glycogen in our models. Importantly, rapamycin increases an effect which is already present in the untreated KO mice. The cause (s) of this effect are presently unknown and warrant further investigation. Interestingly, the reviewer noticed a reduction in CSA in the controls that we commented in the discussion.

Similarly, results state "COX and COX/SDH histochemistry in skeletal muscle showed increased number of COX-positive fibers, and a parallel reduction of SDH hyperintense fibers, another index of mitochondrial proliferation, in treated vs. naïve Cox15sm/sm animals". However, Fig 1B shows persistence of substantial abnormalities in COX and COX/SDH in rapamycin-treated mice. The magnitude of the effect should be quantified and determined if really significantly different from untreated mice in multiple measurements, not just in single image shown.

We quantified COX and SDH staining from n=4/genotype, quantitatively confirming our conclusions. These data are now included in figure 1.

3. The improved motor performance on treadmill test is described as "remarkable", when the Fig 1A shows a significant but very marginal magnitude of effect, where relative to normal baseline (700 m) the mutant mice go from 150-200 m up to 200-250 m at different time points. This is not really remarkable.

As mentioned above, these mice show a severe myopathic phenotype due to the lack of an essential enzyme for the biosynthesis of the hemeA, and the motor performance is severely impaired. After 4 weeks of treatment, KO mice run 83 m vs 172 m of treated KO, i.e. two times as much. Together with the improvement in CSA, these data indicate a rather obvious beneficial effect of rapamycin on the myopathic phenotype. We agree that we are far from the WT performance, but we strongly believe this is a relevant effect. However, we re-worded the sentence to give less emphasis to the in vivo effect.

4. Fig 2A data shows a significant but not complete resolution regarding CS or CIV/CS activities, making any change partial but not "returned to normal values" as stated in results section

In the original manuscript we claimed that CS activity, which was increased in the KO mice, returned to normal values in the treated KO. The reduction in CS activity justify the increase in CIV/CS activity, which remained anyway significantly lower than in WT animals. We clarified this point in the main text.

5. Results and Fig 2B legend states "Protein levels of respiratory chain subunits were reduced in rapamycin-treated vs. naïve Cox15sm/sm mice, to levels comparable to WT (Figure 2B)". However, in the immunoblot shown only UQCRC2 and NDUFA9 levels seem to be less in treated than untreated COX10 mice, and no consistent change with the other 3 complex subunits tested (ATP5A, SDHA, COX4 - where COXIV antibody did not seem to work well in any condition). These data should be replicated and statistical analyses performed on a quantified (ImageJ or otherwise) analysis of replicate data.

We repeated the blots on n=5 samples/genotype, because the standard bands (GAPDH) of some of the samples shown in the blots of the original submission were oversaturated) and may have generated some confusion. We also wish to point out that COXIV antibody gave a low signal in KO muscles because cIV is drastically reduced due to severely impaired COX assembly. The signal becomes almost undetectable in rapamycin-treated animals where mitochondrial content is reduced, as demonstrated by several experiments shown and discussed in the manuscript.

Similarly, the BNG data shown in Fig 1C are not clearly showing a substantial increase in the band marked "cIV", and these need to be replicated and quantified relative to a loading control; in Fig 1D normalized quantitative data from multiple replicate experiments needs to be shown as well to evaluate the magnitude of the effect, which seems marginal at best relative to levels in WT controls.

We agree that in gel activity of cIV was not particularly clear. We removed it from figure 2 because quantified results on cIV activity are shown by the spectrophotometric assay the results of which are displayed in figure 2A.

6. Fig 3 conjectures that mitochondria alterations "possibly result from partially digested organelles within endolysosomes"... but no evidence of this is shown.

We carried out a quantification of disrupted mitochondria by EM which clearly shows a significant reduction of them un rapamycin treated KO muscle samples.

7. The autophagy characterization and flux analyses with in vivo colchicine exposure are well-done and compelling. However, the results conclusion that "These results demonstrate that rapamycin increased the autophagic flux in Cox15sm/sm muscles, and suggest this as a mechanism operating the elimination of dysfunctional mitochondria in the mutant mice" are not substantiated, as at no time do the authors demonstrate "elimination of dysfunctional mitochondria in the mutant mice".

See comment above.

8. Based on the rilmenidine experiments, the authors conclude "autophagy is not the only factor responsible for the rapamycin-mediated effect in Cox15sm/sm mice". As mTORC1 is known to regulate both autophagy and translation, the investigators should also interrogate translation in their animals, which has been shown previously to contribute together with autophagy inhibition to rapamycin effect in mitochondrial disease model animals and cells (PMID: 26041819). However, this citation is not included in the manuscript and no consideration is given to this major mechanistic aspect of mTORC1 inhibition as a therapeutic strategy in mitochondrial disease.

Given the pleiotropic effect of mTOR-signalling, we agree that other mechanisms, including inhibition of translation, can contribute to the overall effect. We checked phospho-S6, a ribosomal protein which is reduced in rapamycin treated animals, suggesting a general reduction of translation. Also, we included a more extensive discussion of this point, as well as the reference suggested by the referee.

These experimental data also appear to be misstated in the conclusion, "Rilmenidine failed to improve the clinical and morphological phenotype, and in fact caused further reduction of fiber size and worsening of the dystrophic lesions of Cox15sm/sm mouse muscles". However, this is not what the data shown in Figs 5C (immunohistochemistry) or 5D (CSA).

Although a trend is suggested by some results, we agree that there is no significant quantitative difference in the muscle phenotype of rilmenidine-treated KO mice compared to untreated KO littermates. We corrected this concept in the revised manuscript.

9. The results in the 3rd section indicate COX10 mutant mice already have increased LAMP1 and TFEB expression, and rapamycin only exacerbates this but does not normalize it at all toward wildtype. In contrast, rilmenidine DOES normalize TFEB expression. Thus, it is not clear how this supports the authors' conclusion that, "These results suggest that sustained lysosomal biogenesis is fundamental to support efficient elimination of dysfunctional mitochondria by autophagy." Isn't this partly obvious, however, since lysosomes are an essential part of the autophagy process, not distinct from it?

Prompted by the comments of the referee we re-evaluated the distribution of TFEB by using immunofluorescence and quantifying the number of nuclei showing co-localization of TFEB and DAPI (instead of using histochemistry, which may give some problems in interpretation). Although the results obtained by IF are not very different from those by IHC, we found that TFEB localization to the nucleus is not significantly different between WT and KO (and rilmenidine), while it is significantly increased in the rapamycin-treated animals of both genotypes. Notably, lysosomes stained with LAMP1 were increased in untreated KO vs WT, probably because of a block in the lysosomal turnover and not because of increased biogenesis, as suggested by the fact that LAMP1 transcript is normal in KO vs WT, while it is increased after rapamycin treatment. These data indicate that lysosomal degradation is blocked or reduced in KO animals, and restored by increased lysosomal biogenesis in the rapa-treated KO. In contrast, rilmenidine does not stimulate lysosomal

biogenesis as it does not induce TFEB translocation to the nucleus. Overall, we think that autophagy and lysosomal biogenesis are, at least in part, two distinct processes.

Further, as above, there is no data showing individual mitochondrial function level and clearance by autophagy to substantiate this claim.

As mentioned above, we substantially expanded our EM analysis to support the conclusion that treated animals had less broken mitochondria as a consequence of increased autophagic flux by rapamycin treatment.

Finally, additional experiments with TFEB knockouts would be needed to support the Conclusion the authors make that "We propose that this effect may be mediated, at least in part, by activation of the Tfeb-dependent transcriptional programme:

Unfortunately, muscle-specific TFEB KO mice develop a myopathy with mitochondrial alterations, and the full body KO is embryonic lethal (Mansueto et al, Cell Metabolism 2016).

10. The final statement in the conclusion is irresponsible and very concerning: "Overall, our data encourage the use of rapamycin or its derivatives in the treatment of mitochondrial diseases." No clinical trial has been performed in human subjects with mitochondrial disease, and this statement should be tempered appropriately to recommend clinical trials to assess efficacy, tolerability and safety (given some of the concerning findings shown in this work), NOT clinical use.

We are obviously not suggesting an indiscriminate use of rapamycin in mitochondrial patients. We do suggest that our data overall support the need for additional studies, eventually leading to clinical application. The reviewer is correct in saying that a large number of clinical data in the target population on safety and tolerability need to be collected before moving to any clinical trial concerning efficacy. We rephrased the sentence accordingly.

Minor Concerns:

1. The references are outdated to describe mitochondrial disease (2004), and tend to self-cite the authors as opposed to giving more updated and comprehensive references to this field that has changed substantially since 2004

We now updated the references.

2. Results section 1 reference growth restriction in Supp Fig S1, but this supplemental file shows mouse ultrastructure only.

We had this in a previous version and left out by mistake. We changed the text correcting it.

3. For the rilmenidine-treatment histochemical analyses shown in Fig 5C, no quantitation is given. The conclusion is that there is no therapeutic effect, but COX/SDH changes shown in Fig 5C with Rilmenidine treatment seem highly similar to what was shown in Fig 1B with rapamycin treatment, except the conclusion in the latter only was that a therapeutic effect was seen. Both should be plotted together on one graph relative to wild-type to fully assess whether any histologic response occurred in either treatment model.

We now quantified all the parameters as we did for rapamycin.

4. Fig 6A - It is not clear what the arrows are pointing too re: "brown staining", as only a portion of the brown staining within nuclei appears to be labelled by white arrows in each panel. Co-staining with a nuclear dye would be helpful to evaluate their statement that rilmenidine did not cause Tfeb translocation to the nucleus.

This experiment has been replaced with the new data obtained by IF.

5. A few spelling and grammatical errors occur throughout, which should be corrected (eg, 'Contrariwise' is not really a word; a "significantly increase" should be "significant increase", gene

names should be italicized and not stated as "gene" after the gene name, etc).

We checked the language carefully and asked a native speaker in our lab to read it.

Referee #3 (Remarks for Author):

The manuscript investigates the effect of long term rapamycin treatment in a mouse model of mitochondrial disease.

The immunohistochemistry presented in Fig 1B is generally convincing, as is the EM imaging. Other sections of the results are not quite so clear cut. Sm/sm +Rap shows less SDH staining than sm/sm -Rap, however the cells do not look especially COX positive, as the DAB staining is weak, and the quantitation is not evaluating the change in COX positive fibres, which would be more convincing.

We quantified COX and SDH staining and confirmed increased Cox-specific reaction, as well as reduced SDH staining, suggesting/confirming reduced mitochondrial content upon rapamycin-treatment.

P8 - The lack of change in CIV activity or indeed relative increase shown in Fig 2A does not quite correspond with the decrease in COX4 by western blot in panel B. Similarly, there is a mild increase in CIV activity in BN gels both in wild type and Cox15sm/sm after rapamycin treatment that does not quite agree with the data in 2B, where COXIV is up in WT +R and down in sm/sm +R. Thus, the text "No differences were detected between treated vs. naïve WT animals" is not a completely accurate reflection of the data as WT+R does show differences in COXIV.

We repeated and quantified the WBs. COXIV behaviour is similar to other OxPhos subunit: it is increased in the KO because of compensatory mitochondrial biogenesis, and is reduced by rapamycin treatment. Overall, our data suggest that KO-rapamycin have less mitochondrial, but the one that are still present work better.

I would suggest that the word 'slightly' or some equivalent be added to "immunovisualization confirmed an 'slightly' increase in COX activity and amount of fully assembled cIV in rapamycin-treated vs. naïve Cox15sm/sm muscle samples (Figure 2C,D)" this would also be more consistent with the phrasing used in the legend.

We changed the text as suggested by the reviewer.

For consistency Fig 3 should have arrows to indicate mitochondria in the WT panel. The size bar is only visible in one panel, for consistency it would be better to be present in all. This is true for other figures.

We updated the figure accordingly.

P9 - the text state "In basal conditions Cox15sm/sm muscles displayed significantly higher LC3-II levels than WT." The significance is not shown on the graphical representation below so it would be more accurate to show the statistical significance or change the wording.

We inserted the asterisks in the figure.

The text states that "Colchicine increased LC3-II in WT but not in Cox15sm/sm muscles.". The increase in WT is very robust but a distinct change also occurs in the sm/sm sample (lane 5 cf lane 6). The levels of LC3-II in both lanes 6 and 8 are higher than lanes 5 and 7. The text describing these changes upon treatment is not entirely consistent with the images presented, which are from single examples of each condition. With mouse experiments it is not uncommon to show that the effect is consistent by showing a number of samples from different animals, as with fig2 and panel A in the same figure. The description of the result would be more convincing if it were seen in multiple

samples.

Since the colchicine alone as well as in combination with rapamycin increased LC3-II levels in sm/sm mice, this reviewer is not convinced by the conclusion that "These results demonstrate that rapamycin increased the autophagic flux in Cox15sm/sm muscles,".

The graph in figure 4C referred to n=3 samples. However, we repeated the blots and added as Figure S3. LC3-II increase in colchicine alone is marginal and not-significant, while the effect of colchicine + rapamycin is significant. We reviewed and made statistical analysis clearer.

Fig 5C has no size bars on the IHC. The fibre size seems to change with +Ril treatment but it is not possible to evaluate this without size bars. The sm/sm +Ril seems to change to have a more varied fibre size. WT H&E in particular looks very different in agreement with the representation in panel D, but all the other WT sections look to have a much smaller CSA than the WT H&E stained section. This does not look consistent across the panel and would be expected to give a larger error bar than seen in panel D for the WT across antibody staining panels.

The bar was reported in the first panel only as it was the same across all the panels. We now put it on all the panels. We carried out the quantification on H&E staining considering 600 fibers/sample, which is a highly representative number.

The legend for Fig 6A states "Note that the increase of the brown staining in the nuclei of rapamycin- but not rilmenidine-treated muscles. Right panel: quantification of n=3 animals/group." This is not evident even when the image is magnified many fold. The data is transformed into a plot to the right of the sections but does not appear to reflect what can be seen in the TFEB staining either in WT or sm/sm samples.

Similarly, the Anti-Lamp1 staining and the quantification are not convincing. Is the quantification looking at number or intensity of signal. All the WT signals are weaker but the decrease in number of foci with rilmenidine looks to be the same in both WT and sm/sm.

We repeated the TFEB localization experiment by using immunofluorescence. The results are rather similar, although we did not confirm increased levels of TFEB in the nuclei of untreated KO vs WT. Accordingly, LAMP1 were normal, suggesting that the increased number of LAMP1 positive vesicles is due to reduced degradation, in keeping with the reduced autophagic flux in KO mice.

The conclusion is that rapamycin could be used for the treatment of all mitochondrial diseases, however, the discussion also stated that rapamycin treatment was "accompanied by inhibition of protein synthesis....", which does not seem like a positive outcome, as it would also induce mitochondrial dysfunction.

Rapamycin has been consistently shown to ameliorate the phenotype of mitochondrial disease models. Our data support a relevant contribution of autophagy to the beneficial effect of rapamycin, but do not exclude contribution from other mTOR-dependent pathways. For instance phospho-S6 is reduced under rapamycin treatment, suggesting a general reduction of translation, as a consequence of mTORC1 inhibition.

The legends frequently repeat the main text and give an interpretation of the result rather than just an explanation of what is being presented. This is not EMM style and is repetitive and in some instance not consistent with the main text.

We revised the legend to avoid repetitions.

Minor points

Some methods are brief. Antibodies should give catalogue numbers.

We changed this accordingly.

P4 - 'protein translation' - should read 'protein synthesis'.

Done.

'skeletal muscle' is written both with and without a hyphen, these should probably be 2 separate words, but should be consistent throughout.

We made it consistent throughout the manuscript.

P13 - muscle spelling needs correcting.

Done

P19 - "into cDNA using XXX kit" the details of the kit used are absent.

We added details.

Some of the phrasing is not very appropriate e.g. 'actually' 3 lines from the end of the discussion

We carefully checked the English throughout the manuscript.

2nd Editorial Decision

20 August 2018

Thank you for the submission of your revised manuscript to EMBO Molecular Medicine. We have now received the enclosed reports from the referees that were asked to re-assess it. As you will see while reviewer 3 is now fully satisfied, referees 1 and 2 still have issues that deserve your attention.

You will see that referee 1 insists that the experiment s/he asked earlier be performed and referee 2 is still concerned by a number of over interpretations that must be fixed. I would like to encourage you to perform the experiment requested by ref. 1 and fix the text in all places alluded to by ref. 2.

I look forward to reading a new revised version of your manuscript as soon as possible.

***** Reviewer's comments *****

Referee #1 (Remarks for Author):

The authors addressed the majority of the issues I had raised in my first revision. The manuscript has been certainly ameliorated. However, I believe that the activation of the Pink1/Parkin cascade remains a key step of the proposed model, and that its precise analysis is of the utmost importance; For this reason, I also believe that Authors should not by any means simply discard my comment and avoid performing the proper experiment with the appropriate reagent. Indeed, a Journal of this caliber deserves this level of accuracy; I thus very strongly suggest them to use the following antibodies that undoubtedly work on mouse tissues: PINK1, from Novus (BC100-494) and Parkin from Abcam (PRK8, ab77924).

Referee #2 (Remarks for Author):

The authors have adequately addressed many of the major concerns about the quantitative data and readjusting to generally more appropriate levels of data interpretations. The grammar is largely better, although a few problematic areas still remain. The area of greatest concern in the revised manuscript is the difficulty following their complex result interpretations, including on several key points stating interpretations of data in the text that appears to be the opposite of the actual results they show in their figures, as detailed below.

1. What does "mitochondrial proficiency" in cells mean, as stated in the revised abstract? This term is not used anywhere in the revised manuscript, and largely muscle tissue histology and motor function seems partially improved with rapamycin. The motor effects remain partial, which should be expanded on in the Discussion for why more substantial improvements in survival and function are not seen despite impressive improvement in tissue histology. Why do

the authors postulate the functional improvements are not greater? Does it relate to timing of therapy, route of administration, tissue penetration, treatment duration, etc? These seem key points to consider if the authors are concluding this should be studied next in human patients.

2. The meaning of "concur to" in the last sentence of the abstract is unclear and awkward. There is no mention of specific results in the abstract that dysfunctional mitochondria are selectively degraded, as opposed to increase seen in functional mitochondria. This makes drawing this conclusion in the last sentence of the abstract difficult for readers to follow or accept.

Further on this point, if there is selective degradation of 'abnormal mitochondria' without change in 'normal mitochondrial levels' as discussed in Results, why don't the authors observe an increase in number of mitochondria "in autophagic vesicles (Fig S2)?"

3. The authors have included new data in results to indicate there is pS6 activation in their model, which is an indicator of mTORC1 signaling attempt to activate cytosolic translation. However, they should recognize and qualify their interpretation to convey that pS6 levels are not alone a direct indicator of translation activity or response. In other words, directly inhibiting translation leads to mTORC1-mediated pS6 increase, but will fail to actually increase translation. For example, measurement of direct translation activity rates in cells has shown that cytosolic translation rates are increased by mitochondrial oxphos inhibition and mildly reduced by low dose (uM range) rapamycin (PMID: 26041819).

The authors stated in their response file that they have included a more complete discussion on the relative contribution of cytosolic translation effects to rapamycin action in the revised manuscript, but actually do not do so anywhere in the Discussion, nor have they included PMID 26041819 reference as they stated in their point by point response file. The Discussion seems unbalanced to focus on modulation of autophagy without translation as a major mechanism of rapamycin action.

Fig 4A is discussed as showing reduction of total S6. However, mTORC1 regulates the level of phosphoS6, not total S6. The authors appear to overinterpret this change in S6, and would need to show additional studies of other ribosome components to be able to reliably conclude it's reflective of altered translation rate in general or just isolated total S6 protein increase. Isotopic incorporation rate studies with deuterated water or S35 Methionine can also be performed if the authors want to draw this conclusion. For these reasons, most investigators only interpret the change in P-S6, not total S6.

4. Regarding discussion of results shown in Fig 2B, the text seems mistaken that COX15 mice ('naïve') had reduced RC subunit expression that further decreased with rapamycin -- rather Fig 2B clearly shows that the untreated mutant mice have complex I/III/V subunits increased relative to wildtype, no change in Complex II subunit, and reduced complex IV cox expression, as expected given their inherent genetic defect synthesizing COX. Fig 2A color bars need a key on the figure itself to readily interpret their meaning.

5. The results discussion on P62 vs LC3-II levels in the Results Fig 4 data interpretation is quite complex and confusing to read. This would benefit by improved interpretation of the authors of how the colchicine experiments, in particular, those showing failure to increase LC3-II levels further indicate impaired autophagy flux in the mutant mice. Also in the discussion, the conclusion about autophagy flux appears based solely on LC3-II levels, without acknowledgement regarding P62 reduction, which seems most clear. Perhaps a schematic interpreting these levels would be helpful since there are many double negatives being reported which makes the interpretation difficult to follow.

a. The logic is not clear how the authors interpret that 'rapamycin alone not changing LC3-II levels' in mutant mice... 'demonstrate rapamycin increased the autophagic flux in Cox15sm/sm muscles'.

b. If their conclusion that "...suggest this as a mechanism operating the elimination of dysfunctional mitochondria in the mutant mice" were true, wouldn't it be expected the histology would show increase mitochondria in autophagosomes, which they report did not occur (in Fig S2)? Rather, rilmenidine clearly is shown in Fig S2 to significantly increase by 2-fold the number of autophagocytosed mitochondria.

c. Figs 4B, C, D are difficult to follow due to lack of key detailing what bar colors or gel columns indicate on the figure itself.

6. Fig 7B and 7C data interpretation are confusing. In the figures, rilmenidine seems to significantly normalize LAMP1 positive vesicles activity (with three asterices shown in Fig 7c) that is elevated in COX15 mutant mice back toward normal while rapamycin treatment only increases it further. Yet, this is surprisingly interpreted by the authors that "rilmenidine had no effect on LAMP1 staining" and "is ineffective" while rapamycin allows for 'sustained lysosome biogenesis'. The more obvious conclusion appears to be the converse, that lysosome biogenesis is induced by mito dysfunction in the COX15 mice, a finding which is not normalized by rapamycin but is normalized completely by the non-mTORC1 dependent autophagy inhibitor rilmenidine. Since the authors are looking to link these data with their earlier finding that muscle physiology is only improved with rapamycin but not rilmenidine, this seems to be overinterpretation of an association the authors are looking for in the data rather than accurate interpretation of the data they generated. This is at best association, rather than confirmation of causation as the authors imply that increased lysosome biogenesis is necessary for improved muscle and mitochondrial function seen with rapamycin.

7. The Discussion is more complete now, with the exception of more balanced discussion of relative role of translation effects of mTORC1 inhibition, as mentioned. However, the concluding paragraph has the 2 last statement both beginning with 'overall': "Overall, our data encourage the use of rapamycin or its derivatives in the treatment of mitochondrial diseases. Overall, our data support the need for additional studies, including clinical trials to test safety, tolerability and eventually efficacy of rapamycin and rapalogs." The first of these remains inappropriate, and should be tempered to explicitly state in animal models, while the second of these statement should clarify further rapamycin clinical study is needed in human mitochondrial disease rather than in general population.

Referee #3 (Remarks for Author):

The authors appear to have addressed comprehensively all the points made by each of the 3 reviewers. The figures have been amended and extended appropriately. The text has also been changed to accommodate a more modest set of interpretations.

SV still appears in the individual contributions even though she has been removed from the author list. This will still need to be amended.

2nd Revision - authors' response

12 September 2018

Referee #1 (Remarks for Author):

The authors addressed the majority of the issues I had raised in my first revision. The manuscript has been certainly ameliorated. However, I believe that the activation of the Pink1/Parkin cascade remains a key step of the proposed model, and that its precise analysis is of the utmost importance; For this reason, I also believe that Authors should not by any means simply discard my comment and avoid performing the proper experiment with the appropriate reagent. Indeed, a Journal of this caliber deserves this level of accuracy; I thus very strongly suggest them to use the following antibodies that undoubtedly work on mouse tissues: PINK1, from Novus (BC100-494) and Parkin from Abcam (PRK8, ab77924).

Referee #2 (Remarks for Author):

The authors have adequately addressed many of the major concerns about the quantitative data and readjusting to generally more appropriate levels of data interpretations. The grammar is largely better, although a few problematic areas still remain. The area of greatest concern in the revised manuscript is the difficulty following their complex result interpretations, including on several key points stating interpretations of data in the text that appears to be the opposite of the actual results they show in their figures, as detailed below.

1. What does "mitochondrial proficiency" in cells mean, as stated in the revised abstract? This term is not used anywhere in the revised manuscript, and largely muscle tissue histology and motor function seems partially improved with rapamycin. The motor effects remain partial, which should be expanded on in the Discussion for why more substantial improvements in survival and function are not seen despite impressive improvement in tissue histology. Why do the authors postulate the functional improvements are not greater? Does it relate to timing of therapy, route of administration, tissue penetration, treatment duration, etc? These seem key points to consider if the authors are concluding this should be studied next in human patients.

We modified the abstract in order to be more specific: "The mTOR inhibitor rapamycin has been reported to ameliorate the clinical and biochemical phenotype of mouse, worm and cellular models of mitochondrial disease, via an unclear mechanism". We also included a discussion on why the effect is limited compared to the amelioration of the muscle phenotype: "In spite of the marked improvement in the muscle morphology and mitochondrial ultrastructure, the improvement in motor performance, albeit significant, is rather limited, and treated mice still perform much less than the untreated littermates. Several reasons may explain this observation, including the age of the mice at the start of treatment, the limited duration of the treatment in our protocol, the intrinsic limitations of our mouse model, which lacks an essential gene, and thus the treatment can only have a limited effect on the biochemical defect".

2. The meaning of "concur to" in the last sentence of the abstract is unclear and awkward. There is no mention of specific results in the abstract that dysfunctional mitochondria are selectively degraded, as opposed to increase seen in functional mitochondria. This makes drawing this conclusion in the last sentence of the abstract difficult for readers to follow or accept.

We specified in the abstract that "Rapamycin treatment restored autophagic flux, which was impaired in naïve *Cox15^{sm/sm}* muscle, and reduced the number of mitochondria with altered morphology, which accumulated in untreated *Cox15^{sm/sm}* mice", and changed concur to contribute.

Further on this point, if there is selective degradation of 'abnormal mitochondria' without change in 'normal mitochondrial levels' as discussed in Results, why don't the authors observe an increase in number of mitochondria "in autophagic vesicles (Fig S2)?

We understand the point the referee wants to make, but we think that the fact that we do not observe an increase of "autophaged mitochondria" in rapamycin treated animals fits with our idea that rapamycin increases the autophagic flux. On the same token, the increase of mitochondria in autophagic vesicles upon rilmenidine simply confirms that mitochondria are delivered to lysosomes (as expected based on the autophagy flux experiment with colchicine), but they are not degraded. We specified this in the discussion. on this point in the text.

3. The authors have included new data in results to indicate there is pS6 activation in their model, which is an indicator of mTORC1 signaling attempt to activate cytosolic translation. However, they should recognize and qualify their interpretation to convey that pS6 levels are not alone a direct indicator of translation activity or response. In other words, directly inhibiting translation leads to mTORC1-mediated pS6 increase, but will fail to actually increase translation. For example, measurement of direct translation activity rates in cells has shown that cytosolic translation rates are increased by mitochondrial oxphos inhibition and mildly reduced by low dose (uM range) rapamycin (PMID: 26041819).

We are not sure we understand this point from the referee, and suspect there is a misunderstanding. We used phosphoS6 as a readout for mTORC1 inhibition, and indeed we observed reduced levels of

phosphoS6 in both wild-type and KO mice after rapamycin. This implies an inhibition (not an activation) of translation, as the referee suggested in his/her original remarks.

The authors stated in their response file that they have included a more complete discussion on the relative contribution of cytosolic translation effects to rapamycin action in the revised manuscript, but actually do not do so anywhere in the Discussion, nor have they included PMID 26041819 reference as they stated in their point by point response file. The Discussion seems unbalanced to focus on modulation of autophagy without translation as a major mechanism of rapamycin action.

We now updated the discussion on this point and included the suggested reference.

Fig 4A is discussed as showing reduction of total S6. However, mTORC1 regulates the level of phosphoS6, not total S6. The authors appear to overinterpret this change in S6, and would need to show additional studies of other ribosome components to be able to reliably conclude it's reflective of altered translation rate in general or just isolated total S6 protein increase. Isotopic incorporation rate studies with deuterated water or S35 Methionine can also be performed if the authors want to draw this conclusion. For these reasons, most investigators only interpret the change in P-S6, not total S6.

We changed the text to make clearer that the translation rate correlates with P-S6, not total S6. We don't have an obvious explanation for increased levels of S6 in *Cox15^{sm/sm}* muscle and for why it is reduced by rapamycin treatment, but this seems to correlate with the severity of the disease.

4. Regarding discussion of results shown in Fig 2B, the text seems mistaken that COX15 mice ('naïve') had reduced RC subunit expression that further decreased with rapamycin -- rather Fig 2B clearly shows that the untreated mutant mice have complex I/III/V subunits increased relative to wildtype, no change in Complex II subunit, and reduced complex IV cox expression, as expected given their inherent genetic defect synthesizing COX. Fig 2A color bars need a key on the figure itself to readily interpret their meaning.

We agree with the reviewer that the sentence was rather confusing and amended it as requested.

5. The results discussion on P62 vs LC3-II levels in the Results Fig 4 data interpretation is quite complex and confusing to read. This would benefit by improved interpretation of the authors of how the colchicine experiments, in particular, those showing failure to increase LC3-II levels further indicate impaired autophagy flux in the mutant mice. Also in the discussion, the conclusion about autophagy flux appears based solely on LC3-II levels, without acknowledgement regarding P62 reduction, which seems most clear. Perhaps a schematic interpreting these levels would be helpful since there are many double negatives being reported which makes the interpretation difficult to follow.

a. The logic is not clear how the authors interpret that 'rapamycin alone not changing LC3-II levels' in mutant mice... 'demonstrate rapamycin increased the autophagic flux in *Cox15^{sm/sm}* muscles'.

We included an additional explanation in the result section: "If rapamycin increases LC3-II synthesis but not its degradation (i.e. it induces autophagic flux), LC3-II should accumulate. If rapamycin induces both synthesis and degradation, LC3-II levels are expected to remain the same.". The fact that LC3-II accumulates after rapamycin+colchicine treatment confirms that autophagic flux is increased.

b. If their conclusion that "...suggest this as a mechanism operating the elimination of dysfunctional mitochondria in the mutant mice" were true, wouldn't it be expected the histology would show increase mitochondria in autophagosomes, which they report did not occur (in Fig S2)? Rather, rilmenidine clearly is shown in Fig S2 to significantly increase by 2-fold the number of autophagocytosed mitochondria.

As explained above, we do not see an increase of mitochondria within the autophagosomes, because rapamycin, but not rilmenidine, increases their degradation. Notably, the experiment with colchicine support the EM data.

c. Figs 4B, C, D are difficult to follow due to lack of key detailing what bar colors or gel columns indicate on the figure itself.

We added the colour codes directly to the graphs.

6. Fig 7B and 7C data interpretation are confusing. In the figures, rilmenidine seems to significantly normalize LAMP1 positive vesicles activity (with three asterices shown in Fig 7c) that is elevated in COX15 mutant mice back toward normal while rapamycin treatment only increases it further. Yet, this is surprisingly interpreted by the authors that "rilmenidine had no effect on LAMP1 staining" and "is ineffective" while rapamycin allows for 'sustained lysosome biogenesis'. The more obvious conclusion appears to be the converse, that lysosome biogenesis is induced by mito dysfunction in the COX15 mice, a finding which is not normalized by rapamycin but is normalized completely by the non-mTORC1 dependent autophagy inhibitor rilmenidine. Since the authors are looking to link these data with their earlier finding that muscle physiology is only improved with rapamycin but not rilmenidine, this seems to be overinterpretation of an association the authors are looking for in the data rather than accurate interpretation of the data they generated. This is at best association, rather than confirmation of causation as the authors imply that increased lysosome biogenesis is necessary for improved muscle and mitochondrial function seen with rapamycin.

Although we agree that we mistakenly commented at Rilmenidine has no effect, while LAMP1-positive vesicles are significantly reduced by rilmenidine treatment, we disagree with the reviewer interpretation for the following reasons. Our EM data clearly show that mitochondria accumulates in autophagosome, but they cannot be degraded. In addition, also damaged mitochondria accumulate in rilmenidine-treated animals, again suggesting that the process of mitochondrial degradation through the lysosomal pathway is impaired, and rilmenidine does not improve it. Second, our data on autophagic flux show that rilmenidine alone reduces the amount of LC3-II and thus the number of autophagosomes. Conversely, colchicine treatment, which blocks autophagy, leads to an accumulation of LC3-II to higher levels than rilmenidine alone. In summary, we have that rilmenidine induces the fusion of autophagosomes with lysosomes (less LC3-II) but the autophagolysosomes cannot degrade their mitochondrial content.

7. The Discussion is more complete now, with the exception of more balanced discussion of relative role of translation effects of mTORC1 inhibition, as mentioned. However, the concluding paragraph has the 2 last statement both beginning with 'overall': "Overall, our data encourage the use of rapamycin or its derivatives in the treatment of mitochondrial diseases. Overall, our data support the need for additional studies, including clinical trials to test safety, tolerability and eventually efficacy of rapamycin and rapalogs." The first of these remains inappropriate, and should be tempered to explicitly state in animal models, while the second of these statement should clarify further rapamycin clinical study is needed in human mitochondrial disease rather than in general population.

This has now been amended.

Referee #3 (Remarks for Author):

The authors appear to have addressed comprehensively all the points made by each of the 3 reviewers. The figures have been amended and extended appropriately. The text has also been changed to accommodate a more modest set of interpretations.

SV still appears in the individual contributions even though she has been removed from the author list. This will still need to be amended.

This has been amended

Corresponding Author Name: Massimo Zeviani
 Journal Submitted to: EMBO Molecular Medicine
 Manuscript Number: EMM-2017-08799-V2